# Benchmarking Positional Encodings for GNNs and Graph Transformers

## Abstract

Recent advances in Graph Neural Networks (GNNs) and Graph Transformers (GTs) have been driven by innovations in architectures and Positional Encodings (PEs), which are critical for augmenting node features and capturing graph topology. PEs are essential for GTs, where topological information would otherwise be lost without message-passing. However, PEs are often tested alongside novel architectures, making it difficult to isolate their effect on established models. To address this, we present a comprehensive benchmark of PEs in a unified framework that includes both message-passing GNNs and GTs. We also establish theoretical connections between MPNNs and GTs and introduce a sparsified GRIT attention mechanism to examine the influence of global connectivity. Our findings demonstrate that previously untested combinations of GNN architectures and PEs can outperform existing methods, offering a more comprehensive picture of the state-of-the-art. To support future research and experimentation in our framework, we make the code publicly available.

## 1 Introduction

Graph machine learning has traditionally relied on message-passing neural networks (MPNNs), which work through iterative rounds of neighborhood aggregation (Kipf & Welling, 2016). In each round, nodes update their states by incorporating information from their neighbors along with their own current states. While effective in capturing local graph structures, this approach can struggle with modeling long-range dependencies. Graph Transformer (GT) architectures utilize full attention mechanisms to circumvent this, but necessitate new methods to integrate graph topology information (Dwivedi & Bresson, 2020). This is similar to how positional encodings (PEs) in Natural Language Processing (NLP) represent token positions within sequences (Vaswani et al., 2017). However, encoding positional information in graphs is more complex than in sequences. Ideally, positional encodings should allow the reconstruction of the graph's topology from node features and provide useful inductive biases to improve performance (Black et al., 2024). Despite the growing number of new graph transformer architectures and positional encodings, there has been a lack of systematic evaluation comparing these encodings across different GT architectures. This makes it difficult to determine whether observed performance improvements are due to novel encodings or architectural innovations.

In this paper, we conduct a comprehensive evaluation of various positional encodings for both message-passing and transformer frameworks. Our goal is to understand the impact of positional encodings on model performance and identify the best combinations of encodings and architectures. By benchmarking state-of-the-art graph transformers with a variety of positional encodings, we provide a clear picture of the current state of the field and offer guidance for future research. Additionally, we further strengthen the theoretical connection between MPNNs and GTs. Although GTs are generally considered fundamentally different due to their use of attention mechanisms, we show that under certain conditions, MPNNs and GTs can be equally expressive, with additional results that extend the scope of previous analyses (Veličković, 2023; Müller & Morris, 2024). Specifically, MPNNs can be applied to fully-connected graphs and operate like a GT, while attention mechanisms can also be adapted for local message-passing. Our theoretical analysis demonstrates that both MPNNs and GTs can have the same expressiveness when the underlying topology of the MPNN is fully connected. Based on these insights, we extend our evaluation to include MPNNs with positional encodings on fully-connected graphs and modify state-of-the-art attention mechanisms for

localized graph convolutions. Our results indicate that by combining existing positional encodings and architectures, state-of-the-art performance can be achieved on several benchmark datasets.

Our contributions can be summarized as follows:

1. We conduct an empirical evaluation of various positional encodings across message-passing neural networks and Graph Transformers to consolidate the current state-of-the-art and find new combinations that surpass the previous best models.

2. We provide theoretical insights into the relationship between MPNNs and GTs, showing conditions where they share similar expressiveness. Based on these observations, we introduce a sparsified version of GRIT attention for localized graph convolutions, which proves effective across multiple datasets.

3. We provide a unified evaluation framework implementing all used architectures and positional encodings in one codebase to facilitate the testing of new positional encodings and models. The code is made publicly available.[1]

## 2 RELATED WORK

**Message-Passing Neural Networks (MPNNs).** Earlier graph neural networks (GNNs), including models like GCN (Kipf & Welling, 2016), GAT (Veličković et al., 2017), GraphSAGE (Hamilton et al., 2017), and GIN (Xu et al., 2018), have paved the way for various advancements. Some convolutional filtering variants incorporate edge attributes into their architecture. GatedGCN (Bresson & Laurent, 2017) employs gates as sparse attention mechanisms, while GINE (Hu et al., 2019) augments features with local edges. Recent efforts aim to enhance the expressive power of GNNs, addressing the limitations imposed by the 1-WL test. For instance, Principal Neighborhood Aggregation (PNA) (Corso et al., 2020) combines different aggregators with degree-scalers to tackle isomorphism tasks in continuous feature spaces. Higher-order GNNs, like k-GNN (Morris et al., 2019; Maron et al., 2019), build on the k-WL algorithm, a more generalized version of the WL test, offering increased expressive power. Other approaches, such as GSN (Bouritsas et al., 2022) and GIN-AK+ (Zhao et al., 2021), utilize substructures (subgraphs) for message passing, while methods like CIN (Bodnar et al., 2021) operate on regular cell complexes, although they remain less powerful than the 3-WL test. Importantly, these models serve as baselines in some graph transformers, demonstrating comparable performance with certain GTs, as cited in GraphGPS (Rampášek et al., 2022), GRIT (Ma et al., 2023), and Exphormer (Shirzad et al., 2023).

**Graph Transformers (GTs).** Graph Transformers (GT) were popularized in recent years (Rampášek et al., 2022; Liu et al., 2023; Mao et al., 2024; Zhang et al., 2023). Modules including positional or structural encodings, global attention, and local message passing are considered as mainstream design components for a standard graph transformer model, which successfully solved the problem of in-scalability (Rampášek et al., 2022; Shirzad et al., 2023) in large graphs, lack of graph inductive bias (Ma et al., 2023), and over-smoothing problems (Chen et al., 2022b). Apart from its maturity in some machine learning fields such as natural language processing, computer vision, or bioinformatics that many previous GT papers have mentioned, GTs have also demonstrated their strength by extending their application to scientific domains such as differential equations (Bryutkin et al., 2024; Choromanski et al., 2022), quantum physics (Wang et al., 2022a), and symbolic regression (Zhong & Meidani). Some recent works are theoretical analysis in graph transformers regarding the theoretical expressive power of GT (Zhou et al., 2024), and the analytical relationship between positional encodings in GT (Keriven & Vaiter, 2024; Black et al., 2024). However, there is currently a lack of a practical benchmark that compares different types of positional encodings. MPNNs and GTs have been compared extensively in the literature, with early work observing that these models can simulate one another (Veličković, 2023). A more rigorous theoretical analysis has demonstrated that GTs can be related to MPNNs when a virtual node is employed (Cai et al., 2023). Furthermore, it has been established that GTs can simulate MPNNs, provided that the positional encodings are sufficiently strong (Müller & Morris, 2024). In contrast, our findings show conditions under which MPNNs operating on fully connected graphs can achieve equal expressiveness to that of GTs, without requiring additional positional encodings or architectural modifications.

---

[1] https://anonymous.4open.science/r/PEGT-34DB

GTs traditionally make use of positional encodings to encode the graph topology, especially when full attention is used. We provide an in-depth review of positional encodings and benchmarking in Section 3.1 and Appendix A.1.

## 3 THEORETICAL FOUNDATIONS

### 3.1 POSITIONAL ENCODINGS

Numerous positional encodings for graph-based models have been discussed in recent research, but they are often scattered across various ablation studies with no unified framework. In this paper, we categorize and streamline the formal definition of existing graph-based positional encodings into three main categories: Laplacian-based, Random walk-based, and others.

We start with some fundamental definitions related to graphs. Let the input graph be $\mathcal{G} = (\mathcal{V}, \mathcal{E}, X)$, where $X \in \mathbb{R}^{|\mathcal{V}|}$ represents the node features. For any graph $\mathcal{G}$, essential properties include the degree matrix $D$ and the adjacency matrix $A$. The graph Laplacian matrix $L$ is defined as $L = D - A$. A normalized graph Laplacian is given by $L = I - D^{-\frac{1}{2}} A D^{-\frac{1}{2}} = U^T \Lambda U$, where the $i$-th row of $U$ corresponds to the graph's $i$-th eigenvector $u_i$, and $\Lambda$ is a diagonal matrix containing the eigenvalues of $L$. We define a graph neural network model $f(\cdot)$ parameterized by $\Theta$. We denote $X_{\text{PE}}^k$ as the positional encoding for node K.

*Laplacian-based* methods utilize functions of the $k$-th eigenvector $U_{k,:}$, $\Lambda$, and parameters $\Theta$. Examples include Laplacian Positional Encoding (**LapPE**) (Rampášek et al., 2022) and Sign-Invariant Networks (**SignNet**) (Lim et al., 2022).

$$X_{\text{PE}}^k = f\left(U_{k,:}, \Lambda, \Theta, \{\cdot\}\right)$$

*Random walk-based* methods are derived from polynomial function $p(\cdot)$ of $D$ and $A$. Examples are Random-Walk Structural Encoding **RWSE** (Rampášek et al., 2022), Random-Walk Diffusion (**RWDIFF / LSPE**) (Dwivedi et al., 2021), and Relative Random Walk Probability Based (**RRWP**) (Ma et al., 2023).

$$X_{\text{PE}}^k = p\left(D, A, \{\cdot\}\right)$$

*Other* methods rely on different procedures, such as colors obtained by mapping 1-WL to higher dimensions. We thus use this umbrella class for all remaining PEs. Examples include the WL-based Positional Encoding (**WLPE**) (Dwivedi & Bresson, 2020) and Graph Convolution Kernel Networks (**GCKN**) (Mialon et al., 2021). We aim to succinctly summarize and unify these positional encoding methods for better accessibility and comparison. The Appendix contains more specific details (including equations) for each positional encoding.

### 3.2 MESSAGE-PASSING NETWORKS

MPNNs comprise multiple layers that repeatedly apply neighborhood aggregation and combine functions to learn a representation vector for each node in the graph. For an input graph $\mathcal{G} = (\mathcal{V}, \mathcal{E}, X)$, the $i$-th layer of a MPNN can be written as

$$c_v^{(i)} = \textbf{COMBINE}^{(i)}\left(c_v^{(i-1)}, \ \textbf{AGGREGATE}^{(i)}\left(\left\{\left\{c_w^{(i-1)} : w \in \mathcal{N}(v)\right\}\right\}\right)\right),$$

where $c_v^{(i-1)}$ represents the state of node $v$ after layer $(i-1)$.

### 3.3 GRAPH TRANSFORMERS

Transformer models have been widely used in modeling sequence-to-sequence data in different domains (Vaswani et al., 2017). Although the attention mechanism has commonly been used to learn on graph-structured data (Veličković et al., 2017), the use of transformers is relatively recent. A GT layer relies on a self-attention module that lets nodes attend to a set of "neighbors", effectively resulting in a dependency graph $G$ of nodes that can attend to each other. We will refer to the nodes that a node $u$ can attend to simply as its neighborhood $\mathcal{N}(u)$. Many architectures use "full attention" on the graph (as opposed to "sparse attention"), meaning that all nodes can attend to all other nodes in the graph, i.e., the underlying dependency graph for attention is fully connected. Based on the

given neighborhood, the generalized attention mechanism first computes attention scores $\alpha_{u,v}$ for every node $u$ and every $v \in \mathcal{N}(u)$, based on the embeddings $c_u^{(i-1)}$ and $c_v^{(i-1)}$ from the previous iteration, and potentially including labels for the edges $(u,v) \in V(G)$. The attention coefficients are then used to weigh the importance of neighbors and compute a new embedding for $u$ as follows:

$$c_u^{(i)} = \Theta \left( c_u^{(i-1)} + \sum_{v \in \mathcal{N}(u)} \alpha_{u,v} \cdot \delta(c_v^{(i-1)}) \right),$$

where $\Theta$ and $\delta$ are transformations for embeddings. This definition aligns with popular architectures such as Exphormer (Shirzad et al., 2023) and GraphGPS (Rampášek et al., 2022). We use Exphormer as a running example to clarify the practical applicability of our proofs, as its attention mechanism can attend to arbitrary neighborhoods. In the case of Exphormer, $\delta$ becomes a linear transformation, $\Theta$ the identity function, and attention coefficients $\alpha_{u,v}$ are computed via dot-product attention that integrates edge labels.

To maintain information about the original topology, adding connectivity information back into the attention mechanism is essential. This is typically done by using positional encodings. Positional encodings can come in the form of node encodings (Rampášek et al., 2022), which are essentially features added to the nodes before the attention block is applied, or edge features, where every edge is endowed with (additional) features, such as the shortest-path-distance between the respective nodes (Ying et al., 2021). In our framework, positional encodings are modeled as labels for nodes in $G$, whereas relative positional encodings can be modeled as edge labels.

### 3.4 Bridging GTs and WL

In the literature, various attempts have been made to bridge the gap between Graph Transformers (GTs) and the WL test (Müller & Morris, 2024; Cai et al., 2023). This is usually done by defining new variants of the WL algorithm that apply to the GT of interest (Kim et al., 2022). However, we argue that such extensions are not necessary. Instead, we can interpret the execution of a GT on $G$ as an MPNN on a new topology $G' = (V, E')$ corresponding to the dependency graph, representing the information flow in the attention layer (Veličković, 2023). For example, a GT with full attention can be seen as an MPNN on the fully connected graph, with $E' = V \times V$. Relative positional encodings can be added to the MPNN as edge labels. This means we can use the (edge-augmented) 1-WL algorithm on $G'$ to upper bound the expressive power of a GT on $G$. While it is perhaps not surprising that GT expressivity can be upper bounded in this way, we also show that GTs can attain this upper bound under some reasonable assumptions. To facilitate this proof, we use the same idea as Xu et al. (2018) to show the equivalence between the GIN architecture and 1-WL.

**Lemma 3.1** (Adapted from Corollary 6 by Xu et al. (2018)). *Assume $\mathcal{X}$ is a countable set. There exists a function $f : \mathcal{X} \to \mathbb{R}^n$ so that for infinitely many choices of $\epsilon$, including all irrational numbers, $h(c, X) = f(c) + \sum_{x \in X} f(x)$ is unique for each pair $(c, X)$, where $c \in \mathcal{X}$ and $X \subseteq \mathcal{X}$ is a multiset of bounded size. Moreover, any function $g$ over such pairs can be decomposed as $g(c, X) = \varphi\left((f(c) + (1 + \epsilon) \sum_{x \in X} f(x)\right)$ for some function $\varphi$.*

See proof on page 16.

To complete the proof for GTs, we adapt Corollary 6 by moving the use of the multiplicative factor $\epsilon \in \mathbb{R}$ from $f(c)$ to the aggregation $\sum_{x \in X} f(x)$. This is because the GT can multiply the aggregation by $\epsilon$ using the attention coefficients while it cannot transform $c_u^{(i-1)}$ directly. The $\epsilon$ is used in the proof to differentiate between embeddings from neighbors and a node's own embedding.

With the adapted Lemma, we can prove the following:

**Theorem 3.2.** *Let $G = (V, E)$ be a graph with node embeddings $c_v$ for nodes $v \in V$. A GT layer on the dependency graph $G' = (V, E')$ can map nodes $v_1, v_2 \in V$ to different embeddings only if the 1-WL algorithm using $E'$ assigns different labels to nodes $v_1$ and $v_2$. For equivalence, we need $\delta$ (in the definition of GTs) to be injective and $\alpha_{u,v} = c$ for a given constant $c \in \mathbb{R}$ and all $(u,v) \in E'$, making the GT as expressive as the 1-WL algorithm.*

See proof on page 17.

The result implies that we can bound the expressiveness of a GT by that of the WL algorithm. As an example, GTs with full attention, as used by Rampášek et al. (2022) and Ma et al. (2023), can be bound by the 1-WL algorithm on the fully connected graph. In this case, we can interpret positional encodings for node pairs as edge features on the complete graph.

In the case of Exphormer, we notice that $\delta$ can be parametrized to be injective when using positional encodings for nodes. This works if the query and key matrices are 0, leading all attention coefficients for a node $u$ to be $\frac{1}{|\mathcal{N}(u)|}$, while the value matrix can be set to the identity matrix times $c$. The only part where Exphormer lacks is the power of $\Theta$, which does not fulfill the requirements in the theorem. Other architectures like GRIT make up for this by using MLPs to encode embeddings (Ma et al., 2023).

We further note that a similar statement can be made for rewiring techniques that change the graph's topology: Applying 1-WL to the rewired topology naturally leads to similar equivalence results. Motivated by the fact that MPNNs and GTs can be seen as applying a "convolution" to some neighborhood, we test how well traditional message-passing convolutions like GatedGCN perform on the fully-connected graph and propose a localized variant of the GRIT attention mechanism that considers a local neighborhood.

### 3.5 SPARSE GRIT MESSAGE-PASSING CONVOLUTION

GRIT introduces two main innovations: (1) A new attention mechanism that updates edge labels on a fully connected graph and (2) RRWP as a positional encoding. While it is relatively easy to use RRWP with both other message-passing and Graph Transformer architectures, we need some adaptions to use the GRIT attention mechanism with message-passing GNNs on sparse graphs. As motivated earlier in Section 3.4, a Graph Transformer can be seen as message-passing on a fully-connected graph. Therefore, we generalize the GRIT attention mechanism designed for fully-connected graphs to a message-passing convolution that works with any neighborhood. We call the resulting convolution *Sparse GRIT*, as it can attend to local neighborhoods on sparse graphs and does not suffer from the quadratic computational overhead that the original GRIT mechanism has. This makes sparse GRIT more efficient and scalable, as we further underline in our empirical evaluation in Section 5.2.

Sparse GRIT utilizes the same updating edge labels $\hat{\mathbf{e}}_{i,j}$ as the original, but only for edges that exist in the original graph. This further distinguishes the convolution from other popular local attention mechanisms like GAT. The main difference to GRIT lies in the update function $\hat{\mathbf{x}}_i$ for nodes, which now attend to their local neighborhood instead of all nodes in the graph. It becomes:

$$\hat{\mathbf{x}}_i = \sum_{j \in \mathcal{N}(i)} \frac{e^{w_j \cdot \hat{\mathbf{e}}_{i,j}}}{\sum_{k \in \mathcal{N}(i)} e^{w_k \cdot \hat{\mathbf{e}}_{i,k}}} \cdot (\mathbf{W}_V \mathbf{x}_j + \mathbf{W}_{E_V} \hat{\mathbf{e}}_{i,j}) \tag{1}$$

where $w_j$ is the attention weight, $\mathbf{W}_V$ and $\mathbf{W}_{E_V}$ are weight matrices. In contrast to GRIT, the summation is taken only over a node's local neighborhood using the implementation of a sparse softmax. With these changes, sparse GRIT works the same as GRIT on a fully connected graph. We, therefore, effectively transform the GRIT GT into an MPNN, which enables us to isolate and analyze what impact the graph that is used for message-passing (fully connected vs. local) has. Results and empirical analysis of the sparse GRIT and GRIT are provided in Section 5.

## 4 BENCHMARKING POSITIONAL ENCODINGS

### 4.1 GENERAL FRAMEWORK

The general GNN framework we consider for our evaluation is depicted in Figure 1. More information on the employed datasets can be found in Appendix A.4. We describe the main components here and give an overview over what methods were tested.

**Design Space 1: Positional Encoding.** As specified in Section 3.1, we test three types of graph-based positional encodings, treating them as node feature augmentations. More background for the different encodings is given in Appendix 3.1

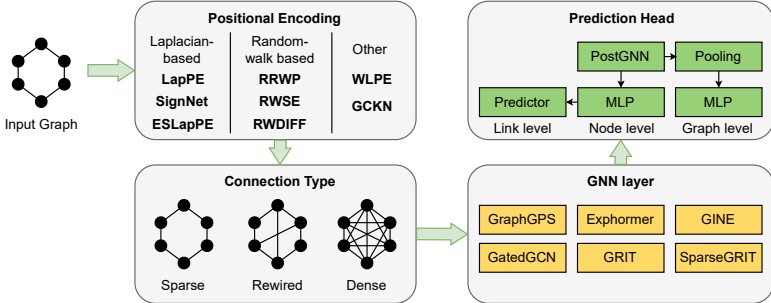

Figure 1: Overview of our evaluation framework, illustrating the preprocessing steps on the left and the GNN model architecture on the right. The framework allows for extensive experimentation with various components, including positional encodings, connection types, and GNN layers. This modular approach facilitates a comprehensive analysis of how different configurations impact model performance. In our experimentation, we mainly focus on the positional encoding and GNN layer, while we also test different connection types.

**Design Space 2: Connection Type.** In most real-world graph datasets, graphs tend to be sparse. This means that message-passing on the original topology can potentially lead to a lack of global information exchange that is necessary for the task. To mitigate this issue, GTs usually employ full-attention on the complete, fully-connected graph (as discussed in Section 3.4). To change the topology that is used for the following GNN layer, we apply a classic MPNN to the fully-connected graph to compare it to GTs and adapt the currently best-performing GT to run on the original graph with our SparseGRIT convolution. While Exphormer implicitly applies some degree of rewiring, we do not consider further rewiring approaches in this work to keep the number of comparisons at a reasonable level.

**Design Space 3: GNN Layers.** Based on the chosen topology, we apply several GNN layers and benchmark their performance. On the MPNN side, we consider GINE, GatedGCN, and SparseGRIT, while we use GraphGPS, Exphormer, and GRIT as classical GTs. The architectures were chosen due to the fact that they are widely used and currently perform best in leaderboards for the tasks we consider. Other convolutions and transformer layers can easily be tested in our general framework.

**Design Space 4: Prediction Heads.** Lastly, we need task-specific prediction heads that decode to either link level, node level, or graph level tasks for the datasets we consider. We use the same setup as popularized by GraphGPS (Rampášek et al., 2022) and do not undertake further testing here.

## 4.2 BENCHMARKING FRAMEWORK

To enable the evaluation of models and future research for measuring the impact of positional encodings, we provide a unified codebase that includes the implementation of all tested models and the respective positional encodings. We base the code off GraphGPS Rampášek et al. (2022) and integrate all missing implementations. This makes for reproducible results and easy extensibility for new datasets, models, or positional encodings. Our codebase further provides readily available implementations for NodeFormer (Wu et al., 2022), Difformer (Wu et al., 2023), GOAT (Kong et al., 2023), GraphTrans (Wu et al., 2021), GraphiT (Mialon et al., 2021), and SAT (Chen et al., 2022a) that are based on the respective original codebases. The code is publicly available at https://anonymous.4open.science/r/PEGT-34DB.

In our experiments, we use five different random seeds for the BENCHMARKINGGNN (Dwivedi et al., 2023) datasets and four for the others. The train-test split settings adhere to those established previously, employing a standard split ratio of 8:1:1. All experiments can be executed on either a single Nvidia RTX 3090 (24GB) or a single RTX A6000 (40GB). To avoid out-of-memory (OOM) issues on LRGB and OGB datasets, we ensure that 100GB of reserved CPU cluster memory is available when pre-transforming positional encodings. Configurations that did not fit into this

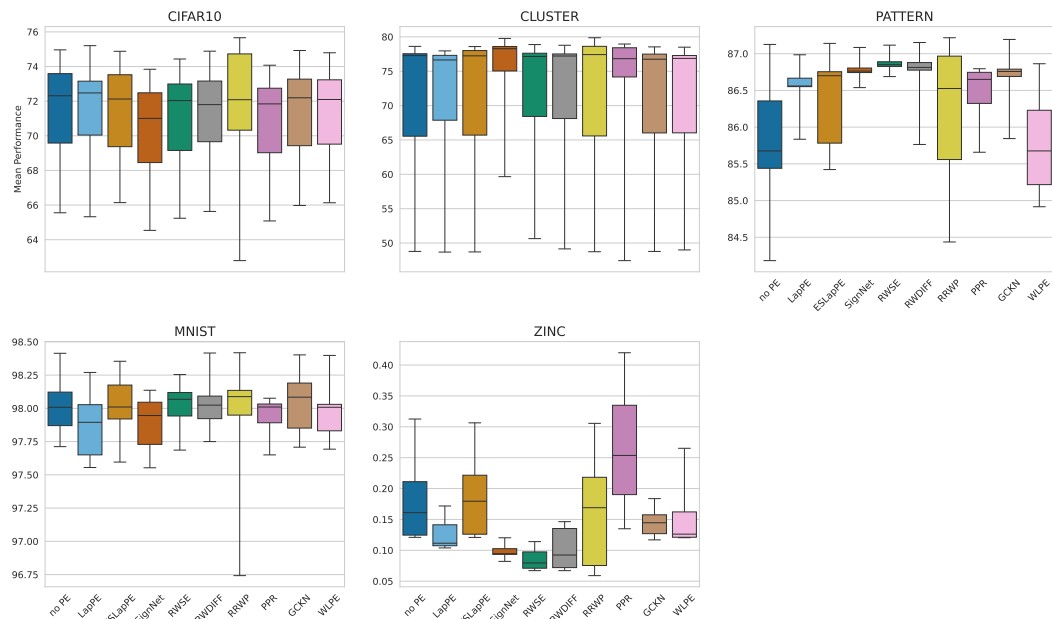

Figure 2: Performance comparison of target metrics across selected datasets from BENCHMARK-INGGNN. The boxplots illustrate the performance range for all models included in the study, with whiskers representing the minimum and maximum performance observed. Notably, RRWP consistently achieves the best results, whereas certain PEs, such as SignNet on CIFAR10, can sometimes decrease performance relative to the baseline without PEs.

computational envelope were not considered. The hyperparameters used for each architecture are provided in the Appendix, as well as running times and memory usage for the PE pre-processing.

## 5 EVALUATION

Based on the framework we established in Section 4.2, we benchmark the performance of different PEs on the BENCHMARKINGGNN (Dwivedi et al., 2023) and LRGB (Dwivedi et al., 2022) datasets. Results for ogbg-molhiv and ogbg-molpcba can be found in Appendix A.7.

### 5.1 BENCHMARKINGGNN DATASETS

We benchmark state-of-the-art models with commonly used PEs in-depth to identify the best configurations. This analysis is often overlooked when new PEs are introduced alongside new architectures without being evaluated with existing models. Our approach decouples the architecture from the PE, allowing us to measure the full range of possible combinations. Our experimental evaluation starts with a dataset-centric approach, examining the effect of various PEs on model performance. Figure 2 illustrates the range of values for the respective target metrics achieved by different PEs. These values are aggregated over all models in our analysis, while more detailed, unaggregated results are available in Appendix A.7. Notably, while we could reproduce most results of previously tested model and PE combinations, we consistently observed slightly worse values for GRIT. This was the case even when using the official codebase and the most up-to-date commit at the time of writing, with provided configuration files intended to reproduce the results stated in the original paper.

Our findings reveal that PEs can significantly influence model performance, with the best choice of PE varying depending on the dataset and task. However, PEs can also negatively impact performance in some cases. For instance, while RRWP performs best on the CIFAR10 dataset and ZINC, there are not always clear winners. Sometimes, good performance can be achieved even without any positional encoding (e.g., for PATTERN). This is also evident when examining the best-performing configurations for each model and PE. While the complete results for all runs are

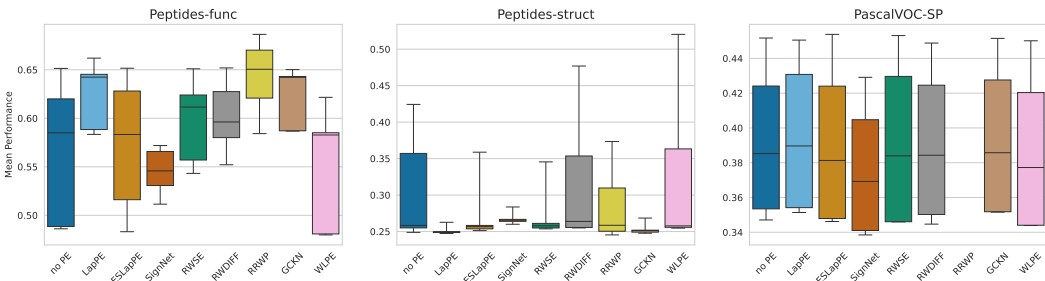

Figure 3: Performance comparison of target metrics across selected datasets from the Long-Range Graph Benchmark. The boxplots illustrate the performance range of all models included in the study, with whiskers indicating the minimum and maximum performance observed. Plots for the remaining datasets are provided in Appendix A.7.

provided in Appendix A.7, we summarize the best-performing configurations for the BENCHMARK-INGGNN datasets in Table 1, indicating which PE led to the best performance for each model and dataset. This enables a fair comparison of all architectures and helps determine the optimal PE overall.

In our comparison, we observe that the sparse GRIT convolution emerges as the best graph convolution for sparse topologies. It competes effectively with the full GRIT attention across most datasets. This suggests that these datasets do not require extensive long-range information exchange and can achieve strong performance with sparse message-passing. The GatedGCN convolution on the fully-connected graph does perform better than the original overall, but generally lacks behind attention-based layers. Regarding the effectiveness of different PEs, random-walk-based encodings such as RRWP and RWSE consistently perform well across the tested models. The only notable exception is the CLUSTER dataset, where SignNet performs competitively for some architectures, although the best results are still achieved with RRWP.

## 5.2 LONG-RANGE GRAPH BENCHMARK

We extend our evaluation to the LRGB datasets and use hyperparameter configurations based on those by Tönshoff et al. (2023), with results presented in Table 2. In these datasets, Laplacian-based encodings generally outperform others (except for the Peptides variations), likely due to their ability to capture more global structure in the slightly larger graphs. This might also be reflected in the fact that transformer-based architectures or models that facilitate global information exchange consistently perform better. Our findings largely align with previous rankings, except for PCQM-Contact, where we achieve a new state-of-the-art with Exphormer, which underscores the importance of thorough benchmarking of existing models. Figure 3 further analyzes the performance of the employed PEs. It is noteworthy that RRWP could not be utilized for larger datasets due to its significant memory footprint and computational complexity, similar to models employing full attention mechanisms. The results align with our previous analysis and show that on datasets like Peptides-func, the PE has

Table 1: Results for the best-performing models and the PE they use for the BENCHMARKINGGNN datasets. All runs except those for EGT and TIGT were done by us. SparseGRIT performs on par with GRIT on most datasets, indicating that full attention might not be necessary for all of them. We color the **best**, **second best**, and **third best** models.

| Model | CIFAR10 ↑ | | CLUSTER ↑ | | MNIST ↑ | | PATTERN ↑ | | ZINC ↓ | |
|---|---|---|---|---|---|---|---|---|---|---|
| EGT (Hussain et al., 2022) | 68.70 ± 0.41 | | 79.23 ± 0.35 | | 98.17 ± 0.09 | | 86.82 ± 0.02 | | 0.108 ± 0.009 | |
| TIGT (Choi et al., 2024) | 73.96 ± 0.36 | | 78.03 ± 0.22 | | 98.23 ± 0.13 | | 86.68 ± 0.06 | | 0.057 ± 0.002 | |
| GINE | 66.14 ± 0.31 | (ESLapSE) | 59.66 ± 0.63 | (SignNet) | 97.75 ± 0.10 | (RWDIFF) | 86.69 ± 0.08 | (RWSE) | 0.075 ± 0.006 | (RWDIFF) |
| GatedGCN | 69.57 ± 0.79 | (RRWP) | 75.29 ± 0.05 | (SignNet) | 97.91 ± 0.08 | (RRWP) | 86.83 ± 0.03 | (RWSE) | 0.102 ± 0.003 | (RWSE) |
| SparseGRIT | 74.95 ± 0.26 | (RRWP) | 79.87 ± 0.08 | (RRWP) | 98.12 ± 0.05 | (RWSE) | 87.17 ± 0.04 | (RRWP) | 0.065 ± 0.003 | (RRWP) |
| Exphormer | 75.21 ± 0.10 | (LapPE) | 78.28 ± 0.21 | (SignNet) | 98.42 ± 0.18 | (RRWP) | 86.82 ± 0.04 | (RWSE) | 0.092 ± 0.007 | (SignNet) |
| GRIT | 75.66 ± 0.41 | (RRWP) | 79.81 ± 0.11 | (RRWP) | 98.12 ± 0.14 | (RRWP) | 87.22 ± 0.03 | (RRWP) | 0.059 ± 0.001 | (RRWP) |
| GatedGCN (FC) | 71.08 ± 0.60 | (RRWP) | 74.78 ± 0.46 | (SignNet) | 98.20 ± 0.15 | (GCKN) | 86.85 ± 0.02 | (RWSE) | 0.114 ± 0.003 | (RWSE) |
| GraphGPS | 72.31 ± 0.20 | (noPE) | 78.31 ± 0.11 | (SignNet) | 98.18 ± 0.12 | (ESLapSE) | 86.87 ± 0.01 | (RWSE) | 0.074 ± 0.006 | (RWSE) |

Table 2: Best-performing models and PEs for the LRGB datasets. We achieve a new state-of-the-art for PCQM-Contact.

| Model | COCO-SP ↑ | | PCQM-Contact ↑ | | PascalVOC-SP ↑ | | Peptides-func ↑ | | Peptides-struct ↓ | |
|---|---|---|---|---|---|---|---|---|---|---|
| GCN (Tönshoff et al., 2023) | $13.38 \pm 0.07$ | | $45.26 \pm 0.06$ | | $0.78 \pm 0.31$ | | $68.60 \pm 0.50$ | | $24.60 \pm 0.07$ | |
| GINE (Tönshoff et al., 2023) | $21.25 \pm 0.09$ | | $46.17 \pm 0.05$ | | $27.18 \pm 0.54$ | | $66.21 \pm 0.67$ | | $24.73 \pm 0.17$ | |
| GatedGCN (Tönshoff et al., 2023) | $29.22 \pm 0.18$ | | $46.70 \pm 0.04$ | | $38.80 \pm 0.40$ | | $67.65 \pm 0.47$ | | $24.77 \pm 0.09$ | |
| CRaWl (Tönshoff et al., 2021) | - | | - | | $45.88 \pm 0.79$ | | $70.74 \pm 0.32$ | | $25.06 \pm 0.22$ | |
| S$^2$GCN (Geisler et al., 2024) | - | | - | | - | | $73.11 \pm 0.66$ | | $24.47 \pm 0.32$ | |
| DRew (Gutteridge et al., 2023) | - | | $34.42 \pm 0.06$ | | $33.14 \pm 0.24$ | | $71.50 \pm 0.44$ | | $25.36 \pm 0.15$ | |
| Graph ViT (He et al., 2023) | - | | - | | - | | $68.76 \pm 0.59$ | | $24.55 \pm 0.27$ | |
| GatedGCN-VN (Rosenbluth et al., 2024) | $32.44 \pm 0.25$ | | - | | $44.77 \pm 1.37$ | | $68.23 \pm 0.69$ | | $24.75 \pm 0.18$ | |
| Exphormer | $34.85 \pm 0.11$ | (ESLapPE) | $47.37 \pm 0.24$ | (LapPE) | $42.42 \pm 0.44$ | (LapPE) | $64.24 \pm 0.63$ | (LapPE) | $24.96 \pm 0.13$ | (LapPE) |
| GraphGPS | $38.91 \pm 0.33$ | (RWSE) | $46.96 \pm 0.17$ | (LapPE) | $45.38 \pm 0.83$ | (ESLapPE) | $66.20 \pm 0.73$ | (LapPE) | $24.97 \pm 0.24$ | (LapPE) |
| SparseGRIT | $19.76 \pm 0.38$ | (noPE) | $45.85 \pm 0.11$ | (LapPE) | $35.19 \pm 0.40$ | (GCKN) | $67.02 \pm 0.80$ | (RRWP) | $24.87 \pm 0.14$ | (LapPE) |
| GRIT | $21.28 \pm 0.08$ | (RWDIFF) | $46.08 \pm 0.07$ | (SignNet) | $35.56 \pm 0.19$ | (noPE) | $68.65 \pm 0.50$ | (RRWP) | $24.54 \pm 0.10$ | (RRWP) |

a consistent impact on the performance, even when the values are aggregated over different architectures. This impact can also be of a negative nature when compared to the baseline that does not use any PE. On other datasets (for example PascalVOC-SP), the PE seems to play a lesser role and good results can be achieved without any PE. The complete results are reported in Appendix A.7.

## 5.3 RUNNING TIME AND MEMORY COMPLEXITY FOR PES

The computational cost of positional encodings (PEs) is a critical consideration, particularly for large graphs where methods with high complexity quickly become infeasible. We evaluated the running time and memory usage for various PEs, and the full results are presented in Appendix A.8.

RRWP is the most memory-intensive PE, but maintains reasonable running times. RWSE and RWDIFF, on the other hand, tend to have significantly longer running times but are relatively more memory-efficient. Laplacian-based methods, such as LapPE and ESLapPE, offer a good balance between computational speed and memory usage, making them practical even for larger datasets. PPR and GCKN come with high computational demands in both time and memory, making them less suited for large-scale graphs. In contrast, Laplacian-based encodings like ESLapPE strike a better trade-off, making them practical for a broader range of graph sizes while still offering competitive performance.

## 5.4 GUIDELINES FOR PRACTICIONERS

For superpixel graph datasets, such as PascalVOC-SP, COCO-SP, MNIST, and CIFAR10, we found that the inclusion of positional encodings generally does not result in substantial performance improvements. In particular, larger superpixel graphs like PascalVOC-SP and COCO-SP showed minimal to no gains from adding PEs, while MNIST similarly exhibited negligible benefits. An exception to this trend is CIFAR10, where RRWP demonstrated potential for enhancing model performance. This suggests that while superpixel graphs may not typically benefit from positional encodings, RRWP could be considered as a candidate for improvement. However, the gains observed may not always justify the increased computational complexity associated with RRWP for such datasets.

In contrast, molecular datasets, such as ZINC and the Peptides variations, displayed a strong dependency on the choice of positional encoding, with significant variations in model performance based on the PE used. For instance, ZINC consistently showed the best results with PPR. On the other hand, the Peptides datasets revealed task-specific preferences: Peptides-func benefited the most from RRWP, while Peptides-struct achieved optimal performance with WLPE. Interestingly, despite using identical graph structures, the two Peptides tasks favored different PEs, which indicates that the nature of the prediction target (functional vs. structural) plays a significant role. Thus, when dealing with molecular datasets, practitioners are advised to experiment with various PEs, as the optimal choice may depend more on the specific task than on the graph structure itself. Still, the optimal PE for a given dataset is generally consistent across different models, which, in combination with the fact that we test on commonly used datasets provides practitioners with a strong starting point for their experiments. This distinction is further highlighted when comparing random-walk-based encodings with Laplacian encodings, where one typically emerges as the clear winner depending on the dataset and task.

# 6 CONCLUSIONS

This study underscores the critical role of positional encodings in enhancing the performance of Graph Neural Networks (GNNs), particularly within Graph Transformer architectures. We conducted a thorough comparison of various positional encodings across a wide array of state-of-the-art models and identify the optimal configurations for diverse datasets, as well as offer valuable insights into the relationship between positional encodings and model performance. While we consolidated much of the current state-of-the-art, we also identified new configurations that surpass the performance of existing best models, such as Exphormer on PCQM-Contact. This underscores the necessity of in-depth comparisons to provide a fair and accurate ranking. Our theoretical considerations have led to the development of the SparseGRIT model. This model shows competitive performance across multiple benchmarks, while maintaining scalability to larger graphs. It shows that sparse message-passing together with the right positional encodings is a viable option on many datasets.

Furthermore, we provide a comprehensive overview of the current state-of-the-art in graph learning and highlight the importance of selecting appropriate positional encodings to achieve optimal results. Our unified codebase includes implementations of all tested models and encodings, which serves as a valuable resource for future research. The framework ensures reproducible results and supports the integration of new datasets, models, and positional encodings, thereby facilitating further experimentation.

**Limitations.** Due to computational constraints, we could not explore all possible hyperparameter configurations and there might be slightly better performing ones that we did not catch. Additionally, although we tested a wide range of models and encodings, it is infeasible to test every model and PE. This is why we focused on current state-of-the-art for both. Further, our evaluations are based on a specific set of benchmark datasets, which may not fully represent the diversity of real-world graph structures. Thus, performance on these benchmarks may not generalize to all types of graph data. Nevertheless, our unified codebase serves as a robust foundation for further testing and development, and enables researchers to overcome these limitations by facilitating the inclusion of new datasets, models, and positional encodings.

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

# A Appendix

## A.1 Extended Related Work

**Positional Encodings for Graphs.** Positional encodings are traditionally used in natural language processing to capture the absolute position of a token within a sentence (Vaswani et al., 2017) or the relative distance between pairs of tokens (Shaw et al., 2018; Ke et al., 2020; Chen, 2021). Similarly, positional encoding in graphs aims to learn both local topology and global structural information of nodes efficiently. This approach has been successfully implemented with the introduction of the graph transformer (Dwivedi & Bresson, 2020). With the advent of graph transformers in the field of graph representation learning, many traditional graph theory methods have been revitalized. Graph signal processing techniques have been employed such as Laplacian decomposition and finite hop random walks (Rampášek et al., 2022; Dwivedi et al., 2023; Ma et al., 2023; Beaini et al., 2021; Dwivedi & Bresson, 2020; Kreuzer et al., 2021; Dwivedi et al., 2021; Lim et al., 2022; Wang et al., 2022b) as absolute or relative positional encoding. Node properties such as degree centrality (Ying et al., 2021) and personalized PageRank (PPR) (Gasteiger et al., 2018; Fu et al., 2024) could be mapped and expanded into higher dimensions for absolute positional encoding, while the shortest distance between nodes could be used for relative positional encoding (Li et al., 2020; Ying et al., 2021). Recent studies have focused on developing learnable positional encodings for graphs (Ying et al., 2021) and exploring their expressiveness and stability as well (Wang et al., 2022b; Ma et al., 2023; Huang et al., 2023). Additionally, graph rewiring combined with layout optimization to coarsen graphs has been proposed as a form of positional encoding (Grötschla et al., 2024).

**GNN Benchmarking.** One of the first GNN benchmarking papers compared architectures with and without positional encodings (PEs) (Dwivedi et al., 2023), where their PE mainly refers to Laplacian positional encoding (LapPE). Their study was limited to the GatedGCN model and discussed the expressive power, robustness, and efficiency of state-of-the-art message-passing methods. Additionally, several surveys have benchmarked the complexity, specific tasks, unified message-passing frameworks (Wu et al., 2020; Zhou et al., 2020), robustness, and privacy (Ju et al., 2024). The LRGB dataset (Dwivedi et al., 2022) has been tested in both GNNs and transformers to demonstrate the superiority of Graph Transformers (GTs) over Message Passing Neural Networks (MPNNs). Many state-of-the-art GTs have included this benchmark in their experiments (Rampášek et al., 2022; Shirzad et al., 2023; Ma et al., 2023). One limitation of the LRGB benchmark is that LRGB only considers LapPE and random walk structural encodings (RWSE). One notable work benchmarked using LRGB by fine-tuning the architectures of GraphGPS and pre-processing (Tönshoff et al., 2023). We adopt their settings but place greater emphasis on the effect of positional encodings.

## A.2 Proofs

**Lemma 3.1** (Adapted from Corollary 6 by Xu et al. (2018)). *Assume $\mathcal{X}$ is a countable set. There exists a function $f : \mathcal{X} \to \mathbb{R}^n$ so that for infinitely many choices of $\epsilon$, including all irrational numbers, $h(c, \mathcal{X}) = f(c) + \sum_{x \in X} f(x)$ is unique for each pair $(c, X)$, where $c \in \mathcal{X}$ and $X \subseteq \mathcal{X}$ is a multiset of bounded size. Moreover, any function $g$ over such pairs can be decomposed as $g(c, X) = \varphi\left((f(c) + (1 + \epsilon) \sum_{x \in X} f(x)\right)$ for some function $\varphi$.*

*Proof of Lemma 3.1.* We slightly tightly follow the proof by Xu et al. (2018) for Corollary 6, but define $h$ as $h(c, X) \equiv f(c) + (1 + \epsilon) \sum_{x \in X} f(x)$ (with $f$ defined as in the original proof). We then want to show that for any $(c', X') \neq (c, X)$ with $c, c' \in \mathcal{X}$ and $X, X' \subset \mathcal{X}, h(c, X) \neq h(c', X')$ holds, if $\epsilon$ is an irrational number. We show the same contradiction as Xu et al. (2018): For any $(c, X)$, suppose there exists $(c', X')$ such that $(c', X') \neq (c, X)$ but $h(c, X) = h(c', X')$ holds. We consider the following two cases: (1) $c' = c$ but $X' \neq X$, and (2) $c' \neq c$. For the first case, $h(c, X) = h(c, X')$ implies $\sum_{x \in X} f(x) = \sum_{x \in X'} f(x)$. By Lemma 5 from Xu et al. (2018) it follows that equality will not hold. For the second case, we can rewrite $h(c, X) = h(c', X')$ as the following equation:

$$\epsilon \cdot \left( \sum_{x \in X} f(x) - \sum_{x \in X'} f(x) \right) = \left( f(c') + \sum_{x \in X'} f(x) \right) - \left( f(c) + \sum_{x \in X} f(x) \right)$$

We assume $\epsilon$ to be irrational, and if $\sum_{x \in X} f(x) - \sum_{x \in X'} f(x) \neq 0$, then the left side of the equation is irrational, while the right side is rational. If $\sum_{x \in X} f(x) - \sum_{x \in X'} f(x) = 0$, then the equation reduces to $f(c') = f(c)$, also a contradiction. $\qquad\square$

**Theorem 3.2.** *Let $G = (V, E)$ be a graph with node embeddings $c_v$ for nodes $v \in V$. A GT layer on the dependency graph $G' = (V, E')$ can map nodes $v_1, v_2 \in V$ to different embeddings only if the 1-WL algorithm using $E'$ assigns different labels to nodes $v_1$ and $v_2$. For equivalence, we need $\delta$ (in the definition of GTs) to be injective and $\alpha_{u,v} = c$ for a given constant $c \in \mathbb{R}$ and all $(u, v) \in E'$, making the GT as expressive as the 1-WL algorithm.*

*Proof of Theorem 3.2.* First, we show that a GT is bounded by 1-WL on the same topology by showing that 1-WL is at least as powerful as a graph transformer. As 1-WL hashes all neighbor states with an injective function, we can observe states from all nodes in the graph in the aggregated multiset at node $v$, including possible edge labels. This information is sufficient to compute the result of the attention module at every node.

For the other direction, we can make use of Lemma 3.1 by setting $c$ to the desired $\epsilon$ and follow the same proof as (Xu et al., 2018). Note that $\Theta$ and $\delta$ have to be powerful enough such that we can apply the universal approximation theorem.

$\qquad\square$

### A.3 TESTED GNN ARCHITECTURES

**Message Passing Neural Networks (MPNN).** For message passing neural networks, we primarily choose `GatedGCN` (Bresson & Laurent, 2017) and `GINE` (Hu et al., 2019). The message-passing update rule for GateGCN is as follows:

$$\mathbf{x}_i^{\ell+1} = f_{\text{G-GCNN}}^{\ell}\left(\mathbf{x}_i^{\ell}, \{\mathbf{x}_j^{\ell} : j \to i\}\right) = \text{ReLU}\left(U^{\ell}\mathbf{x}_i^{\ell} + \sum_{j \to i} \eta_{ij} \odot V^{\ell}\mathbf{x}_j^{\ell}\right) \qquad (2)$$

where $\mathbf{x}_j, j \in \mathcal{N}(i)$ are node features, and $\eta_{ij}$ are edge gates which are employed by $\eta_{ij} = \sigma\left(A^{\ell}x_i^{\ell} + B^{\ell}x_j^{\ell}\right)$. The update for GINE is defined as follows:

$$\mathbf{x}_i' = h_{\Theta}\left((1 + \epsilon) \cdot \mathbf{x}_i + \sum_{j \in \mathcal{N}(i)} \text{ReLU}(\mathbf{x}_j + \mathbf{e}_{i,j})\right) \qquad (3)$$

where $\epsilon$ is a hyper-parameter as specified in GIN paper, edge information $\mathbf{e}_{i,j}$ is injected into individual node features, and MLP $h(\cdot)$ is parameterized by $\Theta$.

Our rationale is as follows:

1. When observing popular MPNNs such as DGN (Beaini et al., 2021), PNA (Corso et al., 2020), and GSN (Bouritsas et al., 2022), they are not consistently scalable or tested on medium-scale datasets as thoroughly as classical MPNNs like `GatedGCN` and `GINE`, as indicated in GraphGPS (Rampášek et al., 2022). This limitation is also evident in current state-of-the-art graph neural networks like CIN (Bodnar et al., 2021) and GIN-AK+ (Zhao et al., 2021), which lack reported results on large-scale graphs, such as most datasets from the Open Graph Benchmark (Hu et al., 2020; Rampášek et al., 2022).

2. Most graph transformers incorporate edge information, making direct comparisons to GNNs without edge information unfair. Graph convolution networks (GCN) (Kipf & Welling, 2016) and Graph Isomorphism Network (GIN) (Xu et al., 2018) potentially lack these updates. Hence, we use modified convolutional graph filters that include edge attributes in message passing, specifically GatedGCN and GINE.

3. We aim to investigate if the results from GatedGCN on fully connected graphs are comparable to those from GT on sparse graphs. In addition to the above points, an improved GatedGCN architecture has been found to perform on par with GT on the peptides-func and peptides-struct datasets (Tönshoff et al., 2023). This finding motivates us to explore the effects of positional encodings on the GatedGCN architecture.

By focusing on GatedGCN and GINE, we aim to leverage their established scalability and performance on medium to large-scale datasets, while fairly comparing their edge attribute capabilities to graph transformers.

**Graph Transformers.** Current Graph Transformers can be divided into two families:

- GTs with only full attention layers. This family includes the following GT(s): EGT (Hussain et al., 2021), GKAT (Choromanski et al., 2022), GRIT (Ma et al., 2023), NAGphormer (Chen et al., 2022b) , Vanilla GT (Dwivedi & Bresson, 2020), GraphiT (Mialon et al., 2021), GRPE (Park et al., 2022), SignNet (Lim et al., 2022), SAN (Kreuzer et al., 2021), Specformer (Bo et al., 2022), TokenGT (Kim et al., 2022), and Transformer-M (Luo et al., 2022a).

- GTs with additional message passing layers as an inductive bias. This family includes the following GT(s): Coarformer (Kuang et al., 2021), Equiformer (Liao & Smidt, 2022), Exphormer (Shirzad et al., 2023), GOAT (Kong et al., 2023), GraphGPS (Rampášek et al., 2022), Graphormer (Ying et al., 2021), GPS++ (Masters et al., 2022), GraphTrans (Wu et al., 2021), SAT (Chen et al., 2022a), NodeFormer (Wu et al., 2022), and URPE (Luo et al., 2022b).

In this research, we select GRIT from the first class, and GraphGPS and Exphormer from the second class. Other models can also be classified into one of these two categories.

**GraphGPS Update.** As specified in the original paper, the model follows a pattern where the output from global attention layers interacts with the output from a global attention (vanilla transformer) layer. In this context, $\mathbf{X}$ represents the node features, $\mathbf{E}$ represents the edge features, and $\mathbf{A}$ is the adjacency matrix.

$$\hat{\mathbf{X}}_M^{\ell+1}, \mathbf{E}^{\ell+1} = \text{MPNN}_e^\ell(\mathbf{X}^\ell, \mathbf{E}^\ell, \mathbf{A}),$$

$$\hat{\mathbf{X}}_T^{\ell+1} = \text{GlobalAttn}^\ell(\mathbf{X}^\ell),$$

$$\mathbf{X}_M^{\ell+1} = \text{BatchNorm}\left(\text{Dropout}\left(\hat{\mathbf{X}}_M^{\ell+1}\right) + \mathbf{X}^\ell\right),$$

$$\mathbf{X}_T^{\ell+1} = \text{BatchNorm}\left(\text{Dropout}\left(\hat{\mathbf{X}}_T^{\ell+1}\right) + \mathbf{X}^\ell\right),$$

$$\mathbf{X}^{\ell+1} = \text{MLP}^\ell\left(\mathbf{X}_M^{\ell+1} + \mathbf{X}_T^{\ell+1}\right)$$

**Exphormer Update.** As specified in the Exphormer paper and observed from its implementation, the model follows a training pattern similar to that used in SAN (Kreuzer et al., 2021):

$$\text{ATTN}_H(\mathbf{X})_{:,i} = \mathbf{x}_i + \sum_{j=1}^{h} \mathbf{W}_O^j \mathbf{W}_V^j \mathbf{X}_{\mathcal{N}_H(i)} \cdot \sigma\left(\left(\mathbf{W}_E^j \mathbf{E}_{\mathcal{N}_H(i)} \odot \mathbf{W}_K^j \mathbf{X}_{\mathcal{N}_H(i)}\right)^T \left(\mathbf{W}_Q^j \mathbf{x}_i\right)\right)$$

where $\mathbf{X}$ is the node features, and $\mathbf{E}$ is the edge features. The most important aspect is that they compute the local sparse attention mechanism using 1) virtual nodes and 2) expander graphs.

## A.4 DATASETS

Statistics and prediction tasks are listed in Table 3. Licenses for each datasets are listed in Table 4.

**BenchmarkingGNN** include *MNIST*, *CIFAR10*, *CLUSTER*, *PATTERN*, and *ZINC*, following the protocols established in *GraphGPS* (Rampášek et al., 2022), *Exphormer* (Shirzad et al., 2023), and *GRIT* (Ma et al., 2023). These datasets have traditionally been employed for benchmarking Graph Neural Networks (GNNs) (Dwivedi et al., 2023), excluding graph transformers. In this paper, we adhere to these established settings but aim to revisit both message passing neural networks (MPNNs) and graph transformers.

**Long-Range Graph Benchmark** (LRGB) (Dwivedi et al., 2022) encompasses *Peptides-func*, *Peptides-struct*, *PascalVOC-SP*, *PCQM-Contact*, and *COCO*. Graph learning in this context is heavily influenced by the interactions between pairs of long-range vertices. Prior research has explored

the potential optimal hyperparameters for both MPNNs and GTs within the LRGB framework (Tönshoff et al., 2023). Our objective is to identify the most effective combination of GNN architectures and positional encoding strategies.

**Open Graph Benchmark** (OGB) (Hu et al., 2020) includes: 1) node-level tasks like *OGBN-Arxiv* and 2) graph-level tasks like *OGBG-MOLHIV* and *OGBG-MOLPCBA*. These datasets are considerably larger in scale compared to the aforementioned benchmarks. Our goal is to discover scalable positional encoding methods, as conventional graph Laplacian decomposition for positional encoding is not feasible for large graphs.

Table 3: Statistics for each dataset

| Dataset | # Graphs | Avg. $|\mathcal{N}|$ | Avg. $|\mathcal{E}|$ | Directed | Prediction level | Prediction task | Metric |
|---|---|---|---|---|---|---|---|
| ZINC | 12,000 | 23.2 | 24.9 | No | graph | regression | Mean Abs. Error |
| MNIST | 70,000 | 70.0 | 564.5 | Yes | graph | 10-class classif. | Accuracy |
| CIFAR10 | 60,000 | 117.6 | 941.1 | Yes | graph | 10-class classif. | Accuracy |
| PATTERN | 14,000 | 118.9 | 2,359.2 | No | inductive node | binary classif. | Accuracy |
| CLUSTER | 12,000 | 117.2 | 1,510.9 | No | inductive node | 6-class classif. | Accuracy |
| PascalVOC-SP | 11,355 | 479.4 | 2,710.5 | No | inductive node | 21-class classif. | F1 score |
| COCO-SP | 123,286 | 476.4 | 2,693.7 | No | inductive node | 81-class classif. | F1 score |
| PCQM-Contact | 529,434 | 30.1 | 69.1 | No | inductive link | link ranking | MRR (Fil.) |
| Peptides-func | 15,535 | 150.9 | 307.3 | No | graph | 10-task classif. | Avg. Precision |
| Peptides-struct | 15,535 | 150.9 | 307.3 | No | graph | 11-task regression | Mean Abs. Error |
| ogbn-arxiv | 1 | 169,343 | 1,166,243 | Yes | transductive node | 40-class classif. | Accuracy |
| ogbg-molhiv | 41,127 | 25.5 | 27.5 | No | graph | binary classif. | AUROC |
| ogbg-molpcba | 437,929 | 26.0 | 28.1 | No | graph | 128-task classif. | Avg. Precision |

In future work, we hope to add more large-scale inductive datasets such as OGBG-PPA, OGBG-Code2 and PCQM4Mv2 (Hu et al., 2020), and transductive datasets such as CS, Physics and Computer and Photo (Shirzad et al., 2023) into comparison.

## A.5 POSITIONAL ENCODINGS

### A.5.1 LAPLACIAN BASED METHODS

We define $L$ as the Laplacian matrix for our input graph $\mathcal{G} = (\mathcal{V}, \mathcal{E})$. According to graph theory, as it's positive semidefinite and symmetric, it could be further decomposed as $L = \sum_i \lambda_i u_i u_i^T$, where $\lambda_i$ is the eigenvalue and $u_i$ is the eigenvector. Under a unified scheme of positional encoding for graph neural networks, we define a normalized graph Laplacian $L = I - D^{-\frac{1}{2}} A D^{-\frac{1}{2}} = U^T \Lambda U$ where i-th row of U corresponds to the graph's i-th eigenvector $u_i$, and $\Lambda$ is a diagonal matrix containing all eigenvalues. Under the Laplacian-based settings, we could express each positional encoding for node k in a similar way by:

Table 4: Dataset licenses.

| Dataset | License |
|---|---|
| ZINC | MIT License |
| MNIST | CC BY-SA 3.0 and MIT License |
| CIFAR10 | CC BY-SA 3.0 and MIT License |
| PATTERN | MIT License |
| CLUSTER | MIT License |
| PascalVOC-SP | Custom license and CC BY 4.0 License |
| COCO-SP | Custom license and CC BY 4.0 License |
| PCQM-Contact | CC BY 4.0 License |
| Peptides-func | CC BY-NC 4.0 License |
| Peptides-struct | CC BY-NC 4.0 License |
| ogbn-arxiv | MIT License |
| ogbg-molhiv | MIT License |
| ogbg-molpcba | MIT License |

$$X_{\text{PE}}^k = \mathbf{f}(U_{k,:}, \Lambda, \Theta, \{\cdot\}) \tag{4}$$

where $U_{k,:}$ represents the i-th row of U, $\Lambda$ is a diagonal matrix containing all eigenvalues, $\Theta$ is the function parameters which represent the linear or non-linear operations on $U$ and $\Lambda$, and $\{\cdot\}$ is the additional parameters that are utilized by each method individually. We consider three Laplacian-based methods: Laplacian Positional Encoding (*LapPE*), Sign-Invariant Positional Encoding (*SignNet*), and rectified Graph Convolution Kernel Network-based Positional Encoding (*GCKN*).

**LapPE (Rampášek et al., 2022)** LapPE, or Laplacian Positional Encoding, is a method that leverages the eigenvectors of the graph Laplacian to encode positional information for nodes. The core idea is that the eigenvectors corresponding to higher eigenvalues contain more information about the local structure of the graph, especially the relationships between a node and its neighbors. We can further concatenate the eigenvectors with their corresponding eigenvalue. In the actual implementation, an additional parameter $\mathcal{S}$ is employed to randomly split the sign of this concatenated

eigenvector. Subsequently, we apply either a DeepSet (Zaheer et al., 2017) or an MLP $\Phi$ parameterized by $\Theta$ to this eigenvector.

$$X_{\text{PE}}^k = \mathbf{f}(U_{k,:}, \Lambda, \Theta, \mathcal{S}) \tag{5}$$

$$= \Phi_\Theta\left(\mathcal{S} \odot (U_{k,:} \parallel \Lambda_k)\right) \tag{6}$$

The equivariant and stable version of LapPE (ESLapPE) (Wang et al., 2022b) follows the same procedure but omits the post-processing MLP.

**SignNet (Lim et al., 2022)** SignNet is an advanced version of LapPE, which considers both the original eigenvector and its inversely signed counterparts. An additional graph neural network (**GNN**) is applied to capture local Laplacian signals before passing them to the MLP. The outputs from two distinct or shared GNNs are then added together. This approach is proven to be sign-invariant and capable of approximating any continuous function of eigenvectors with the desired invariances (Lim et al., 2022). While we did not consider BasisNet in this work, researchers could further explore its inclusion in their studies for comparison with SignNet. The expression for SignNet is:

$$X_{\text{PE}}^k = \mathbf{f}(U_{i,:}, \Lambda, \Theta, A) \tag{7}$$

$$= \Phi_\Theta\left(\mathbf{GNN}(U_{k,:} \parallel \Lambda_k, A) + \mathbf{GNN}((-\mathbf{1}) \odot (U_{k,:} \parallel \Lambda_k), A)\right) \tag{8}$$

**GCKN (Mialon et al., 2021)** According to GraphiT (Mialon et al., 2021), there are two ways to construct the new graph Laplacian matrix, either by using diffusion kernels (GCKN) or a p-step random walk kernel (p-RWSE). Here, we introduce the diffusion kernel (p-step is similar) where the new Laplacian is computed by multiplying by inverse $\beta$ and then placed onto the exponential of e. $U$ is the new eigenvector matrix. The method is similar to LapPE, but the additional parameter $\beta$ is used to control the diffusion process (Kondor & Vert, 2004). The expression for GCKN is given by:

$$X_{\text{PE}}^k = \mathbf{f}(U_{k,:}, \Lambda, \Theta, \{\mathcal{S}, \beta\}) \tag{9}$$

$$= \Phi_\Theta\left(\mathcal{S} \odot (U_{k,:} \parallel \Lambda_k)\right), \text{where } U_k^T \Lambda U_k = e^{-\beta L} \tag{10}$$

### A.5.2 RANDOM WALK BASED METHODS

We denote $p$ as a polynomial function. From the following settings, we can see that this class of methods takes a polynomial of $D^{-1}A$, which is:

$$X_{\text{PE}}^k = p(D, A, \{\cdot\}) \tag{11}$$

**RWSE (Rampášek et al., 2022)** Random Walk Structural Encoding (RWSE) encodes the graph structure by computing the frequencies of random walks starting from each node. Specifically, the RWSE method calculates the probabilities of nodes being visited at each step of the random walk. This approach utilizes the polynomial $(D^{-1}A)^k$, where $D$ is the degree matrix and $A$ is the adjacency matrix, to represent the result of a $k$-step random walk. For the biased version, weights $\theta_k$ are used to weight the results of each step. The formula is as follows:

$$X_{\text{PE}}^k = p(D, A, K) \tag{12}$$

$$= \sum_{k=1}^K (D^{-1}A)^k, \text{or} = \sum_{k=1}^K \theta_k (D^{-1}A)^k \text{ if biased} \tag{13}$$

**RWDIFF (LSPE) (Dwivedi et al., 2021)** Learnable positional encoding, on the other hand, can encode positions through random walk diffusion and decouples structural and positional encoding (Dwivedi et al., 2021). Unlike RWSE, RWDIFF concatenates the random walk diffusion features from each time step (new dimension), while RWSE directly adds those k-step random walk matrices. The initial condition where no random walk is performed is also considered, with the additional parameter $I$, which is the identity matrix. The formula is as follows:

$$X_{\text{PE}}^k = p(D, A, \{K, I\}) \tag{14}$$

$$= [I, D^{-1}A, (D^{-1}A)^2, (D^{-1}A)^3, ..., (D^{-1}A)^{K-1}]_{k,k} \tag{15}$$

$$= I_{k,k} \parallel (D^{-1}A)_{k,k} \parallel (D^{-1}A)_{k,k}^2 \parallel (D^{-1}A)_{k,k}^3 \parallel ... \parallel (D^{-1}A)_{k,k}^{K-1} \tag{16}$$

**RRWP (Ma et al., 2023)**  One improvement on RWDIFF in GRIT (Ma et al., 2023) is that the sparse graph is connected to a fully connected graph for each graph in the batch, as well as an addition update in the edge features where it has shown a better performance on LRGB benchmark. It is stated that it is at least as expressive as a biased RWSE (Ma et al., 2023). The form of $X_{\text{PE}}^k$ is not changed here, instead we mention the edge attributes (as the structural encoding). The edge feature is also indicated by random walk diffusion, however, the off-diagonal entry indicates the edge features, which is represented by a form of probability from node i to j:

$$P_{i,j} = [I, D^{-1}A, (D^{-1}A)^2, (D^{-1}A)^3, ..., (D^{-1}A)^{K-1}]_{i,j} \tag{17}$$

$$= I_{i,j} \parallel (D^{-1}A)_{i,j} \parallel (D^{-1}A)_{i,j}^2 \parallel (D^{-1}A)_{i,j}^3 \parallel ... \parallel (D^{-1}A)_{i,j}^{K-1} \tag{18}$$

**PPR (Gasteiger et al., 2018)**  Personalized PageRank (PPR) propagation is an approximate and faster propagation scheme under message passing (Gasteiger et al., 2018). For each node, its PageRank is given in its analytical form as:

$$X_{\text{PE}}^k = p(D, A, \{\alpha, |\mathcal{V}|\}) \tag{19}$$

$$= \alpha \left( \mathbb{I}_{|\mathcal{V}|} - (1-\alpha)D^{-1}A \right)^{-1} i_k \tag{20}$$

where $i_k$ is the indicator function, and $\alpha$ controls the distance from the root node. It is considered one of the positional encodings in GRIT (Ma et al., 2023), which is strictly less powerful than RRWP. From its analytical solution, it is also classified under random walk-based methods since the function is inversely related to $D^{-1}A$.

### A.5.3  OTHER METHODS

**WLPE (Dwivedi & Bresson, 2020)**  The Weisfeiler-Lehman Positional Encoding (WLPE) method, as introduced by Dwivedi & Bresson (2020), leverages the Weisfeiler-Lehman (WL) graph isomorphism test to generate positional encodings for nodes in a graph. Firstly, the hashed node feature for node k $X_k'$ is updated by using a hash function that combines the node's own feature $X_k$ with the features of its neighbors:

$$X_k' = \text{hash}\left(X_k, \{X_u : u \in N(v), v \in \mathcal{V}\}\right) \tag{21}$$

Here, $N(v)$ denotes the neighborhood of node $v$, and $\mathcal{V}$ is the set of all nodes in the graph. Secondly, the positional encoding $X_{\text{PE}}^k$ is generated by applying a function $f$ to $X_k'$ and the hidden dimension $d_h$:

$$X_{\text{PE}}^k = f(X_k', d_h)$$

The function $f$ typically involves sinusoidal transformations to embed the positional information into a continuous vector space. This transformation is detailed as follows:

$$X_{\text{PE}}^k = \left[ \sin\left( \frac{X_k'}{10000^{\frac{2l}{d_h}}} \right), \cos\left( \frac{X_k'}{10000^{\frac{2l+1}{d_h}}} \right) \right]_{l=0}^{\left\lfloor \frac{d_h}{2} \right\rfloor}$$

In this expression:

- $X_k'$ is the hashed feature of node $k$.
- $d_h$ is the hidden dimension, controlling the size of the positional encoding.
- $l$ ranges from 0 to $\left\lfloor \frac{d_h}{2} \right\rfloor$, ensuring that the resulting vector has $d_h$ dimensions.

### A.6  MODEL CONFIGURATIONS

We provide the model configuration here to ensure reproducibility.

**BENCHMARKINGGNN**  For BENCHMARKINGGNN, we adhere to established settings from relevant literature for each model. Specifically, for the GatedGCN and GraphGPS models, we follow the configurations detailed in the GraphGPS paper (Rampášek et al., 2022). For the Exphormer model, we utilize the settings from the Exphormer paper (Shirzad et al., 2023). For the GINE, Sparse GRIT, and Global GRIT models, we adopt the configurations from the GRIT paper (Ma et al., 2023). We provide five tables, one for each dataset, to ensure comprehensive coverage of the BENCHMARKINGGNN. Unlike the GraphGPS paper, which fixed the positional encoding, we will report the statistics of the computation of positional encoding in separate tables. Configurations are listed from table 5 to table 9.

Table 5: Model Configurations for MNIST

| Model | lr | dropout | heads | layers | hidden dim | pooling | batch size | epochs | norm | PE dim |
|---|---|---|---|---|---|---|---|---|---|---|
| GatedGCN | 0.001 | 0.0 | - | 4 | 52 | mean | 16 | 100 | - | 8 |
| GINE | 0.001 | 0.0 | - | 3 | 52 | mean | 16 | 150 | BatchNorm | 18 |
| GraphGPS | 0.001 | 0.0 | 4 | 3 | 52 | mean | 16 | 100 | BatchNorm | 18 |
| Exphormer | 0.001 | 0.1 | 4 | 5 | 40 | mean | 16 | 150 | BatchNorm | 8 |
| GRITSparseConv | 0.001 | 0.0 | - | 3 | 52 | mean | 16 | 150 | BatchNorm | 18 |
| GRIT | 0.001 | 0.0 | 4 | 3 | 52 | mean | 16 | 150 | BatchNorm | 18 |

Table 6: Model Configurations for CIFAR10

| Model | lr | dropout | heads | layers | hidden dim | pooling | batch size | epochs | norm | PE dim |
|---|---|---|---|---|---|---|---|---|---|---|
| GatedGCN | 0.001 | 0.0 | - | 4 | 52 | mean | 16 | 100 | - | 8 |
| GINE | 0.001 | 0.0 | - | 4 | 52 | mean | 16 | 150 | BatchNorm | 18 |
| GraphGPS | 0.001 | 0.0 | 4 | 3 | 52 | mean | 16 | 150 | BatchNorm | 8 |
| Exphormer | 0.001 | 0.1 | 4 | 5 | 40 | mean | 16 | 150 | BatchNorm | 8 |
| GRITSparseConv | 0.001 | 0.0 | - | 3 | 52 | mean | 16 | 150 | BatchNorm | 18 |
| GRIT | 0.001 | 0.0 | 4 | 3 | 52 | mean | 16 | 150 | BatchNorm | 18 |

Table 7: Model Configurations for PATTERN

| Model | lr | dropout | heads | layers | hidden dim | pooling | batch size | epochs | norm | PE dim |
|---|---|---|---|---|---|---|---|---|---|---|
| GatedGCN | 0.001 | 0.0 | - | 4 | 64 | - | 32 | 100 | BatchNorm | 10 |
| GINE | 0.0005 | 0.0 | - | 10 | 64 | - | 32 | 100 | BatchNorm | 21 |
| GraphGPS | 0.001 | 0.0 | 4 | 6 | 64 | - | 32 | 100 | BatchNorm | 10 |
| Exphormer | 0.0002 | 0.0 | 4 | 4 | 40 | - | 32 | 100 | BatchNorm | 10 |
| GRITSparseConv | 0.0005 | 0.0 | - | 8 | 64 | - | 32 | 100 | BatchNorm | 21 |
| GRIT | 0.0005 | 0.0 | 8 | 10 | 64 | - | 32 | 100 | BatchNorm | 21 |

Table 8: Model Configurations for CLUSTER

| Model | lr | dropout | heads | layers | hidden dim | pooling | batch size | epochs | norm | PE dim |
|---|---|---|---|---|---|---|---|---|---|---|
| GatedGCN | 0.001 | 0.0 | - | 4 | 48 | - | 16 | 100 | BatchNorm | 16 |
| GINE | 0.001 | 0.01 | - | 2 | 64 | - | 16 | 100 | BatchNorm | 16 |
| GraphGPS | 0.001 | 0.01 | 8 | 16 | 48 | - | 16 | 100 | BatchNorm | 16 |
| Exphormer | 0.0002 | 0.1 | 8 | 20 | 32 | - | 16 | 200 | BatchNorm | 8 |
| SparseGRIT | 0.001 | 0.01 | - | 16 | 48 | - | 16 | 100 | BatchNorm | 32 |
| GRIT | 0.0005 | 0.0 | 8 | 10 | 64 | - | 32 | 100 | BatchNorm | 21 |

Table 9: Model Configurations for ZINC

| Model | lr | dropout | heads | layers | hidden dim | pooling | batch size | epochs | norm | PE dim |
|---|---|---|---|---|---|---|---|---|---|---|
| GatedGCN | 0.001 | 0.0 | - | 4 | 64 | add | 32 | 1700 | BatchNorm | 8 |
| GINE | 0.001 | 0.0 | - | 10 | 64 | add | 32 | 1500 | BatchNorm | 21 |
| GraphGPS | 0.001 | 0.0 | 4 | 10 | 64 | add | 32 | 1500 | BatchNorm | 21 |
| Exphormer | 0.001 | 0.0 | 4 | 4 | 64 | add | 32 | 1500 | BatchNorm | 8 |
| SparseGRIT | 0.001 | 0.0 | - | 10 | 64 | add | 32 | 1500 | BatchNorm | 21 |
| GRIT | 0.001 | 0.0 | 8 | 10 | 64 | add | 32 | 1500 | BatchNorm | 18 |

**Long Range Graph Benchmark** We mainly consider four models: GatedGCN, GraphGPS, Exphormer, and Sparse GRIT. For GatedGCN and GraphGPS, we primarily follow the fine-tuned configurations as described by Tonshoff et al. (2023) (Tönshoff et al., 2023). For Sparse GRIT, we adopt the hyperparameters used for the peptides-func and peptides-struct datasets and transfer these settings to the COCO-SP, Pascal-VOC, and PCQM-Contact datasets, as detailed by Dwivedi et al. (2022) (Dwivedi et al., 2022). For Exphormer, we follow the configurations proposed by Shirzad et al. (2023) (Shirzad et al., 2023). Configurations are listed from table 10 to table 14.

**Open Graph Benchmark** We mainly consider three models: GraphGPS, Exphormer, and Sparse GRIT. Due to scalability issues, we do not include configurations for the GPS model for ogbn-arxiv. GraphGPS (Rampášek et al., 2022), Exphormer (Shirzad et al., 2023) and Sparse GRIT have shared the same settings for the ogbg-molpcba dataset. Configurations are listed from table 15 to table 17.

Table 10: Model Configurations for Peptides-func

| Model | lr | dropout | heads | layers | hidden dim | pooling | batch size | epochs | norm | PE dim |
|---|---|---|---|---|---|---|---|---|---|---|
| GatedGCN | 0.001 | 0.1 | - | 10 | 95 | mean | 200 | 250 | BatchNorm | 16 |
| GraphGPS | 0.001 | 0.1 | 4 | 6 | 76 | mean | 200 | 250 | BatchNorm | 16 |
| Exphormer | 0.0003 | 0.12 | 4 | 8 | 64 | mean | 128 | 200 | BatchNorm | 16 |
| GRITSparseConv | 0.0003 | 0.0 | - | 4 | 96 | mean | 16 | 300 | BatchNorm | 16 |
| GRIT | 0.0003 | 0.0 | 4 | 4 | 96 | mean | 16 | 200 | BatchNorm | 17 |

Table 11: Model Configurations for Peptides-struct

| Model | lr | dropout | heads | layers | hidden dim | pooling | batch size | epochs | norm | PE dim |
|---|---|---|---|---|---|---|---|---|---|---|
| GatedGCN | 0.001 | 0.1 | - | 8 | 100 | mean | 128 | 250 | BatchNorm | 16 |
| GraphGPS | 0.001 | 0.1 | 4 | 8 | 64 | mean | 200 | 250 | BatchNorm | 16 |
| Exphormer | 0.0003 | 0.12 | 4 | 4 | 88 | mean | 128 | 200 | BatchNorm | 16 |
| GRITSparseConv | 0.0003 | 0.05 | - | 4 | 96 | mean | 16 | 300 | BatchNorm | 16 |
| GRIT | 0.0003 | 0.0 | 8 | 4 | 96 | mean | 32 | 200 | BatchNorm | 24 |

Table 12: Model Configurations for PCQM-Contact

| Model | lr | dropout | heads | layers | hidden dim | pooling | batch size | epochs | norm | PE dim |
|---|---|---|---|---|---|---|---|---|---|---|
| GatedGCN | 0.001 | 0.1 | - | 8 | 215 | - | 500 | 100 | - | 16 |
| GraphGPS | 0.001 | 0.0 | 4 | 6 | 76 | - | 500 | 150 | BatchNorm | 16 |
| Exphormer | 0.0003 | 0.0 | 4 | 7 | 64 | - | 128 | 200 | BatchNorm | 16 |
| GRITSparseConv | 0.001 | 0.0 | - | 10 | 64 | - | 500 | 100 | BatchNorm | 16 |

Table 13: Model Configurations for Pascal-VOC

| Model | lr | dropout | heads | layers | hidden dim | pooling | batch size | epochs | norm | PE dim |
|---|---|---|---|---|---|---|---|---|---|---|
| GatedGCN | 0.001 | 0.2 | - | 10 | 95 | - | 50 | 200 | - | 16 |
| GraphGPS | 0.001 | 0.1 | 4 | 8 | 68 | - | 50 | 200 | BatchNorm | 16 |
| Exphormer | 0.0005 | 0.15 | 8 | 4 | 96 | - | 32 | 300 | BatchNorm | 16 |
| GRITSparseConv | 0.001 | 0.0 | - | 10 | 64 | - | 50 | 250 | BatchNorm | 16 |

Table 14: Model Configurations for COCO-SP

| Model | lr | dropout | heads | layers | hidden dim | pooling | batch size | epochs | norm | PE dim |
|---|---|---|---|---|---|---|---|---|---|---|
| GatedGCN | 0.001 | 0.1 | - | 6 | 120 | - | 16 | 200 | - | 16 |
| GraphGPS | 0.001 | 0.1 | 4 | 8 | 68 | - | 50 | 200 | BatchNorm | 16 |
| Exphormer | 0.0005 | 0.0 | 4 | 7 | 72 | - | 32 | 200 | BatchNorm | 16 |
| GRITSparseConv | 0.0005 | 0.0 | - | 4 | 64 | - | 32 | 200 | BatchNorm | 16 |

Table 15: Model Configurations for OGBN-Arxiv

| Model | lr | dropout | heads | layers | hidden dim | pooling | batch size | epochs | norm | PE dim |
|---|---|---|---|---|---|---|---|---|---|---|
| Exphormer | 0.001 | 0.3 | 2 | 4 | 80 | add | 1 | 600 | BatchNorm | 16 |
| GRITSparseConv | 0.001 | 0.1 | - | 4 | 64 | add | 1 | 600 | BatchNorm | 8 |

Table 16: Model Configurations for OGBG-Molhiv

| Model | lr | dropout | heads | layers | hidden dim | pooling | batch size | epochs | norm | PE dim |
|---|---|---|---|---|---|---|---|---|---|---|
| GraphGPS | 0.0001 | 0.05 | 4 | 10 | 64 | mean | 32 | 100 | BatchNorm | 8 |
| Exphormer | 0.0001 | 0.05 | 4 | 8 | 64 | mean | 32 | 100 | BatchNorm | 16 |
| GRITSparseConv | 0.0001 | 0.0 | - | 8 | 64 | mean | 32 | 100 | BatchNorm | 16 |

Table 17: Model Configurations for OGBG-Molpcba

| Model | lr | dropout | heads | layers | hidden dim | pooling | batch size | epochs | norm | PE dim |
|---|---|---|---|---|---|---|---|---|---|---|
| GraphGPS | 0.0005 | 0.2 | 4 | 5 | 384 | mean | 512 | 100 | BatchNorm | 20 |
| Exphormer | 0.0005 | 0.2 | 4 | 5 | 384 | mean | 512 | 100 | BatchNorm | 20 |
| GRITSparseConv | 0.0005 | 0.2 | - | 5 | 384 | mean | 512 | 100 | BatchNorm | 20 |

### A.7 COMPLETE RESULTS

This section provides the results of all runs that we conducted in the paper. This also includes the non-aggregated results that show the performance of every positional encoding on any model and dataset.

#### A.7.1 BENCHMARKINGGNN

Table 18 has listed all results of GNN models with different positional encodings on BENCHMARKINGGNN datasets. Figure 4 shows a percentage of improvement compared to GNN models without any positional encoding.

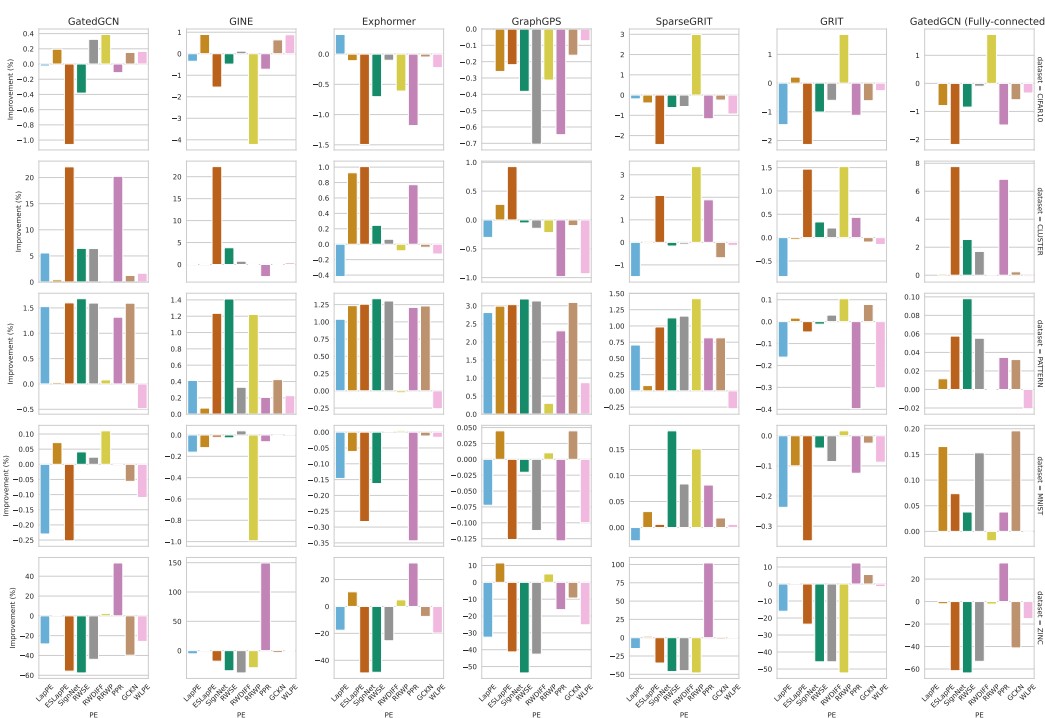

Figure 4: Percentage of improvement compared to GNN models without any positional encoding (BENCHMARKINGGNN)

#### A.7.2 LONG RANGE GRAPH BENCHMARK

Table 19 has listed all results of GNN models with different positional encodings on BENCHMARKINGGNN dataset. Figure 5 shows a percentage of improvement compared to GNN models without any positional encoding. Figure 6 shows the mean average of improvement on the Long Range Graph Benchmark for each positional encoding individually.

#### A.7.3 OPEN GRAPH BENCHMARK

Table 20 has listed all results of GNN models with different positional encodings on Open Graph Benchmark dataset.

### A.8 STATISTICS FOR POSITIONAL ENCODINGS

We measure both the time that is taken to measure pre-computing positional enoodings (PEs), as well as the space that CPU is taken to precompute it, which are presented from Table 21 to Table 26. Figure 7 shows the comparison between time and memory for each dataset, which is log-scaled.

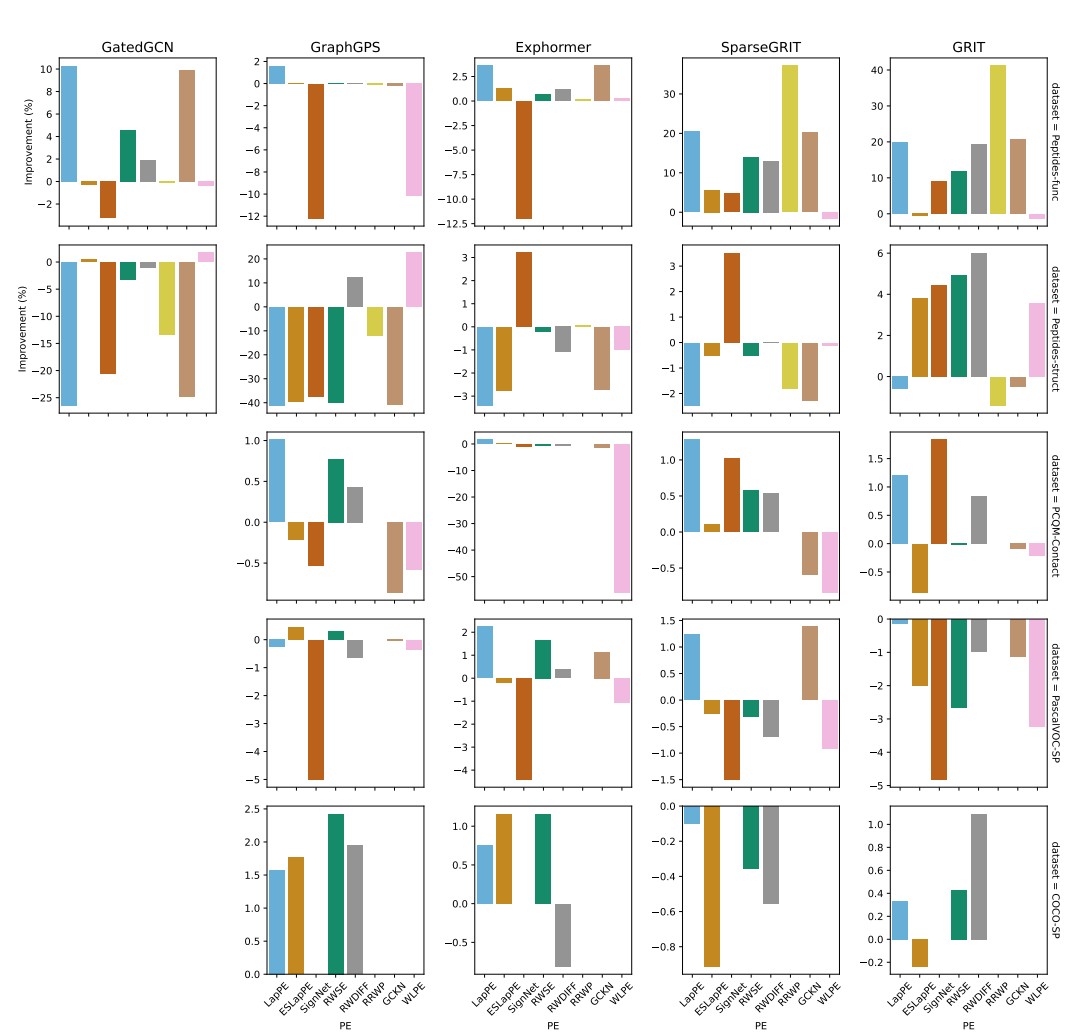

Figure 5: Percentage of improvement compared to GNN models without any positional encoding (LRGb)

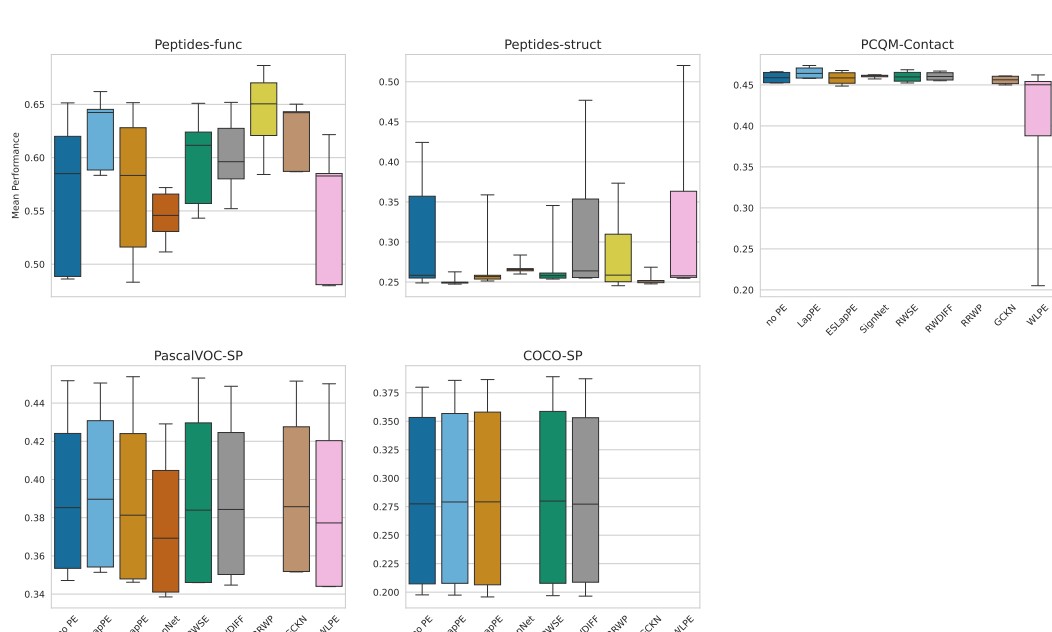

Figure 6: Mean performance of different positional encodings on Long Range Graph Benchmark

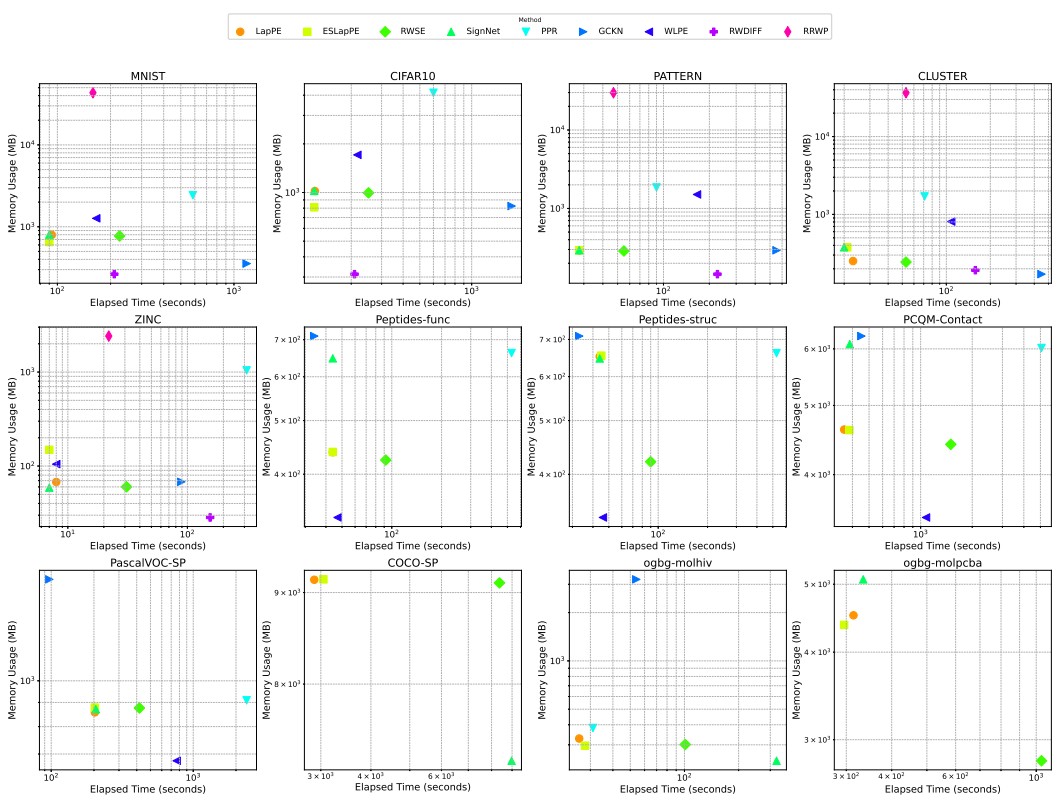

Figure 7: Temporal complexity vs. spatial complexity for different positional encodings

Table 18: Different positional encodings with GNNs on BENCHMARKINGGNN including ZINC, MNIST, CIFAR10, PATTERN, and CLUSTER. Experiments are run on a NVIDIA RTX 3090 and RTX A6000. Five random seeds are: 0, 7, 42, 100, and 2024 (although it should be noted that the execution of PyG on the cuda backend is non-deterministic). Note that the batched graphs are sparse as default. Batched graphs are only fully-connected when it comes to RRWP.

| Sparse Graph | MNIST ↑ | CIFAR10 ↑ | PATTERN ↑ | CLUSTER ↑ | ZINC ↓ |
|---|---|---|---|---|---|
| GatedGCN + noPE | $97.800_{\pm 0.138}$ | $69.303_{\pm 0.318}$ | $85.397_{\pm 0.040}$ | $61.695_{\pm 0.261}$ | $0.2398_{\pm 0.0094}$ |
| GatedGCN + ESLapPE | $97.870_{\pm 0.090}$ | $69.438_{\pm 0.297}$ | $85.422_{\pm 0.161}$ | $61.953_{\pm 0.082}$ | $0.2409_{\pm 0.0131}$ |
| GatedGCN + LapPE | $97.575_{\pm 0.025}$ | $69.285_{\pm 0.205}$ | $86.700_{\pm 0.000}$ | $65.130_{\pm 0.405}$ | $0.1718_{\pm 0.0024}$ |
| GatedGCN + RWSE | $97.840_{\pm 0.171}$ | $69.038_{\pm 0.152}$ | $86.833_{\pm 0.030}$ | $65.675_{\pm 0.296}$ | $0.1016_{\pm 0.0030}$ |
| GatedGCN + SignNet | $97.553_{\pm 0.167}$ | $68.570_{\pm 0.240}$ | $86.763_{\pm 0.027}$ | $75.293_{\pm 0.047}$ | $0.1060_{\pm 0.0021}$ |
| GatedGCN + PPR | $97.797_{\pm 0.045}$ | $69.224_{\pm 0.546}$ | $86.522_{\pm 0.093}$ | $74.175_{\pm 0.122}$ | $0.3678_{\pm 0.0198}$ |
| GatedGCN + GCKN | $97.745_{\pm 0.069}$ | $69.408_{\pm 0.222}$ | $86.758_{\pm 0.049}$ | $62.478_{\pm 0.156}$ | $0.1446_{\pm 0.0048}$ |
| GatedGCN + WLPE | $97.693_{\pm 0.235}$ | $69.418_{\pm 0.165}$ | $84.980_{\pm 0.160}$ | $62.738_{\pm 0.291}$ | $0.1779_{\pm 0.0059}$ |
| GatedGCN + RWDIFF | $97.823_{\pm 0.119}$ | $69.528_{\pm 0.494}$ | $86.760_{\pm 0.043}$ | $65.653_{\pm 0.470}$ | $0.1346_{\pm 0.0074}$ |
| GatedGCN + RRWP | $97.908_{\pm 0.076}$ | $69.572_{\pm 0.787}$ | $85.465_{\pm 0.148}$ | $61.728_{\pm 0.174}$ | $0.2451_{\pm 0.0131}$ |
| GINE + noPE | $97.712_{\pm 0.120}$ | $65.554_{\pm 0.225}$ | $85.482_{\pm 0.272}$ | $48.783_{\pm 0.060}$ | $0.1210_{\pm 0.0107}$ |
| GINE + ESLapPE | $97.596_{\pm 0.071}$ | $66.140_{\pm 0.310}$ | $85.546_{\pm 0.114}$ | $48.708_{\pm 0.061}$ | $0.1209_{\pm 0.0066}$ |
| GINE + LapPE | $97.555_{\pm 0.045}$ | $65.325_{\pm 0.195}$ | $85.835_{\pm 0.195}$ | $48.685_{\pm 0.035}$ | $0.1144_{\pm 0.0028}$ |
| GINE + RWSE | $97.686_{\pm 0.073}$ | $65.238_{\pm 0.283}$ | $86.688_{\pm 0.084}$ | $50.642_{\pm 0.694}$ | $0.0795_{\pm 0.0034}$ |
| GINE + SignNet | $97.692_{\pm 0.165}$ | $64.538_{\pm 0.314}$ | $86.538_{\pm 0.044}$ | $59.660_{\pm 0.630}$ | $0.0993_{\pm 0.0069}$ |
| GINE + PPR | $97.650_{\pm 0.088}$ | $65.082_{\pm 0.434}$ | $85.658_{\pm 0.048}$ | $47.440_{\pm 2.290}$ | $0.3019_{\pm 0.0122}$ |
| GINE + GCKN | $97.708_{\pm 0.105}$ | $65.976_{\pm 0.308}$ | $85.844_{\pm 0.157}$ | $48.780_{\pm 0.149}$ | $0.1169_{\pm 0.0029}$ |
| GINE + WLPE | $97.716_{\pm 0.118}$ | $66.132_{\pm 0.225}$ | $85.676_{\pm 0.084}$ | $48.997_{\pm 0.068}$ | $0.1205_{\pm 0.0062}$ |
| GINE + RWDIFF | $97.750_{\pm 0.097}$ | $65.632_{\pm 0.553}$ | $85.764_{\pm 0.209}$ | $49.148_{\pm 0.168}$ | $0.0750_{\pm 0.0058}$ |
| GINE + RRWP | $96.742_{\pm 0.277}$ | $62.790_{\pm 1.501}$ | $86.526_{\pm 0.036}$ | $48.736_{\pm 0.108}$ | $0.0857_{\pm 0.0009}$ |
| Exphormer + noPE | $98.414_{\pm 0.047}$ | $74.962_{\pm 0.631}$ | $85.676_{\pm 0.049}$ | $77.500_{\pm 0.151}$ | $0.1825_{\pm 0.0209}$ |
| Exphormer + ESLapPE | $98.354_{\pm 0.108}$ | $74.880_{\pm 0.322}$ | $86.734_{\pm 0.024}$ | $78.218_{\pm 0.267}$ | $0.2023_{\pm 0.0140}$ |
| Exphormer + LapPE | $98.270_{\pm 0.070}$ | $75.205_{\pm 0.095}$ | $86.565_{\pm 0.075}$ | $77.175_{\pm 0.165}$ | $0.1503_{\pm 0.0117}$ |
| Exphormer + RWSE | $98.254_{\pm 0.084}$ | $74.434_{\pm 0.205}$ | $86.820_{\pm 0.040}$ | $77.690_{\pm 0.147}$ | $0.0933_{\pm 0.0050}$ |
| Exphormer + SignNet | $98.136_{\pm 0.094}$ | $73.842_{\pm 0.317}$ | $86.752_{\pm 0.088}$ | $78.280_{\pm 0.211}$ | $0.0924_{\pm 0.0072}$ |
| Exphormer + PPR | $98.076_{\pm 0.126}$ | $74.076_{\pm 0.104}$ | $86.712_{\pm 0.047}$ | $78.098_{\pm 0.211}$ | $0.2414_{\pm 0.0123}$ |
| Exphormer + GCKN | $98.402_{\pm 0.067}$ | $74.926_{\pm 0.288}$ | $86.730_{\pm 0.040}$ | $77.470_{\pm 0.067}$ | $0.1690_{\pm 0.0056}$ |
| Exphormer + WLPE | $98.398_{\pm 0.162}$ | $74.794_{\pm 0.358}$ | $85.454_{\pm 0.033}$ | $77.402_{\pm 0.120}$ | $0.1465_{\pm 0.0095}$ |
| Exphormer + RWDIFF | $98.416_{\pm 0.055}$ | $74.886_{\pm 0.810}$ | $86.792_{\pm 0.023}$ | $77.550_{\pm 0.057}$ | $0.1360_{\pm 0.0082}$ |
| Exphormer + RRWP | $98.418_{\pm 0.179}$ | $74.504_{\pm 0.369}$ | $85.652_{\pm 0.001}$ | $77.434_{\pm 0.056}$ | $0.1914_{\pm 0.0153}$ |
| GraphGPS + noPE | $98.136_{\pm 0.085}$ | $72.310_{\pm 0.198}$ | $84.182_{\pm 0.276}$ | $77.590_{\pm 0.158}$ | $0.1610_{\pm 0.0045}$ |
| GraphGPS + ESLapPE | $98.180_{\pm 0.117}$ | $72.122_{\pm 0.511}$ | $86.700_{\pm 0.055}$ | $77.800_{\pm 0.107}$ | $0.1795_{\pm 0.0110}$ |
| GraphGPS + LapPE | $98.065_{\pm 0.075}$ | $72.310_{\pm 0.530}$ | $86.550_{\pm 0.150}$ | $77.355_{\pm 0.115}$ | $0.1086_{\pm 0.0062}$ |
| GraphGPS + RWSE | $98.116_{\pm 0.102}$ | $72.034_{\pm 0.756}$ | $86.866_{\pm 0.010}$ | $77.550_{\pm 0.195}$ | $0.0744_{\pm 0.0060}$ |
| GraphGPS + SignNet | $98.012_{\pm 0.091}$ | $72.152_{\pm 0.323}$ | $86.734_{\pm 0.069}$ | $78.308_{\pm 0.111}$ | $0.0945_{\pm 0.0019}$ |
| GraphGPS + PPR | $98.010_{\pm 0.097}$ | $71.842_{\pm 0.325}$ | $86.124_{\pm 0.214}$ | $76.828_{\pm 0.250}$ | $0.1349_{\pm 0.0054}$ |
| GraphGPS + GCKN | $98.180_{\pm 0.117}$ | $72.194_{\pm 0.515}$ | $86.786_{\pm 0.043}$ | $77.514_{\pm 0.182}$ | $0.1460_{\pm 0.0078}$ |
| GraphGPS + WLPE | $98.038_{\pm 0.134}$ | $72.258_{\pm 0.661}$ | $84.916_{\pm 0.195}$ | $76.866_{\pm 0.171}$ | $0.1204_{\pm 0.0055}$ |
| GraphGPS + RWDIFF | $98.026_{\pm 0.101}$ | $71.800_{\pm 0.363}$ | $86.820_{\pm 0.063}$ | $77.478_{\pm 0.150}$ | $0.0924_{\pm 0.0212}$ |
| GraphGPS + RRWP | $98.146_{\pm 0.105}$ | $72.084_{\pm 0.466}$ | $84.436_{\pm 0.224}$ | $77.420_{\pm 0.080}$ | $0.1690_{\pm 0.0084}$ |
| SparseGRIT + noPE | $97.940_{\pm 0.071}$ | $72.778_{\pm 0.627}$ | $85.948_{\pm 0.148}$ | $77.274_{\pm 0.170}$ | $0.1255_{\pm 0.0062}$ |
| SparseGRIT + ESLapPE | $97.970_{\pm 0.110}$ | $72.494_{\pm 0.501}$ | $86.018_{\pm 0.319}$ | $77.238_{\pm 0.066}$ | $0.1280_{\pm 0.0077}$ |
| SparseGRIT + LapPE | $97.915_{\pm 0.065}$ | $72.640_{\pm 0.040}$ | $86.555_{\pm 0.025}$ | $76.100_{\pm 0.085}$ | $0.1070_{\pm 0.0017}$ |
| SparseGRIT + RWSE | $98.122_{\pm 0.054}$ | $72.330_{\pm 0.600}$ | $86.914_{\pm 0.031}$ | $77.148_{\pm 0.174}$ | $0.0676_{\pm 0.0060}$ |
| SparseGRIT + SignNet | $97.946_{\pm 0.122}$ | $71.003_{\pm 0.301}$ | $86.794_{\pm 0.055}$ | $78.882_{\pm 0.146}$ | $0.0821_{\pm 0.0043}$ |
| SparseGRIT + PPR | $98.020_{\pm 0.194}$ | $71.926_{\pm 0.833}$ | $86.650_{\pm 0.033}$ | $78.732_{\pm 0.202}$ | $0.2536_{\pm 0.0193}$ |
| SparseGRIT + GCKN | $97.958_{\pm 0.127}$ | $72.598_{\pm 0.535}$ | $86.650_{\pm 0.033}$ | $76.746_{\pm 0.187}$ | $0.1233_{\pm 0.0071}$ |
| SparseGRIT + WLPE | $97.946_{\pm 0.125}$ | $72.096_{\pm 0.835}$ | $85.712_{\pm 0.027}$ | $77.170_{\pm 0.143}$ | $0.1262_{\pm 0.0059}$ |
| SparseGRIT + RWDIFF | $98.022_{\pm 0.083}$ | $72.366_{\pm 0.388}$ | $86.938_{\pm 0.045}$ | $77.214_{\pm 0.065}$ | $0.0690_{\pm 0.0039}$ |
| SparseGRIT + RRWP | $98.088_{\pm 0.048}$ | $74.954_{\pm 0.256}$ | $87.168_{\pm 0.041}$ | $79.872_{\pm 0.079}$ | $0.0651_{\pm 0.0027}$ |
| GRIT + noPE | $98.108_{\pm 0.190}$ | $74.402_{\pm 0.135}$ | $87.126_{\pm 0.033}$ | $78.616_{\pm 0.178}$ | $0.1237_{\pm 0.0057}$ |
| GRIT + ESLapPE | $98.010_{\pm 0.141}$ | $74.558_{\pm 0.682}$ | $87.140_{\pm 0.064}$ | $78.588_{\pm 0.111}$ | $0.1241_{\pm 0.0031}$ |
| GRIT + LapPE | $97.875_{\pm 0.001}$ | $73.325_{\pm 0.505}$ | $86.985_{\pm 0.015}$ | $77.960_{\pm 0.310}$ | $0.1039_{\pm 0.0035}$ |
| GRIT + RWSE | $98.068_{\pm 0.182}$ | $73.652_{\pm 0.623}$ | $87.116_{\pm 0.046}$ | $78.880_{\pm 0.057}$ | $0.0671_{\pm 0.0037}$ |
| GRIT + SignNet | $97.766_{\pm 0.220}$ | $72.812_{\pm 0.482}$ | $87.085_{\pm 0.064}$ | $79.770_{\pm 0.150}$ | $0.0945_{\pm 0.0098}$ |
| GRIT + PPR | $97.986_{\pm 0.082}$ | $73.568_{\pm 0.451}$ | $86.780_{\pm 0.001}$ | $78.958_{\pm 0.175}$ | $0.1390_{\pm 0.0076}$ |
| GRIT + GCKN | $98.084_{\pm 0.139}$ | $73.946_{\pm 0.910}$ | $87.194_{\pm 0.044}$ | $78.542_{\pm 0.149}$ | $0.1306_{\pm 0.0141}$ |
| GRIT + WLPE | $98.022_{\pm 0.173}$ | $74.206_{\pm 0.684}$ | $86.863_{\pm 0.033}$ | $78.500_{\pm 0.091}$ | $0.1218_{\pm 0.0035}$ |
| GRIT + RWDIFF | $98.024_{\pm 0.148}$ | $73.956_{\pm 0.202}$ | $87.152_{\pm 0.045}$ | $78.778_{\pm 0.090}$ | $0.0671_{\pm 0.0060}$ |
| GRIT + RRWP | $98.124_{\pm 0.141}$ | $75.662_{\pm 0.410}$ | $87.217_{\pm 0.034}$ | $79.812_{\pm 0.109}$ | $0.0590_{\pm 0.0010}$ |

Table 19: Five Long Range Graph Benchamrk Datasets which include Peptides_func, Pepteide_struct, PCQM_Contact, PascalVOC-SuperPixels and COCO-SuperPixels. The hyper-parameters for Peptides_func and Pepteide_struct follow the original GraphGPS settings. For GraphGPS, we follow the settings of a special study into LRGB (Tönshoff et al., 2023) where it found a state-of-the-art settings for GraphGPS on those five datasets.

| Sparse Graph | Peptides-func | Peptides-struct | PCQM-Contact | PascalVOC-SP | COCO-SP |
|---|---|---|---|---|---|
| GatedGCN + noPE | $0.6523_{\pm 0.0074}$ | $0.2470_{\pm 0.0005}$ | $0.4730_{\pm 0.0003}$ | $0.3923_{\pm 0.0020}$ | $0.2619_{\pm 0.0045}$ |
| GatedGCN + LapPE | $0.6581_{\pm 0.0068}$ | $0.2472_{\pm 0.0003}$ | $0.4764_{\pm 0.0004}$ | $0.3920_{\pm 0.0033}$ | $0.2671_{\pm 0.0006}$ |
| GatedGCN + ESLapPE | $0.6484_{\pm 0.0037}$ | $0.2490_{\pm 0.0020}$ | $0.4736_{\pm 0.0006}$ | $0.3930_{\pm 0.0041}$ | $0.2628_{\pm 0.0004}$ |
| GatedGCN + RWSE | $0.6696_{\pm 0.0022}$ | $0.2485_{\pm 0.0022}$ | $0.4749_{\pm 0.0005}$ | $0.3882_{\pm 0.0041}$ | $0.2657_{\pm 0.0007}$ |
| GatedGCN + SignNet | $0.5327_{\pm 0.0137}$ | $0.2688_{\pm 0.0016}$ | $0.4672_{\pm 0.0001}$ | $0.3814_{\pm 0.0005}$ | - |
| GatedGCN + GCKN | $0.6544_{\pm 0.0040}$ | $0.2483_{\pm 0.0009}$ | $0.4687_{\pm 0.0002}$ | $0.3933_{\pm 0.0044}$ | - |
| GatedGCN + WLPE | $0.6562_{\pm 0.0053}$ | $0.2473_{\pm 0.0012}$ | $0.4671_{\pm 0.0003}$ | $0.3805_{\pm 0.0018}$ | - |
| GatedGCN + RWDIFF | $0.6527_{\pm 0.0053}$ | $0.2474_{\pm 0.0003}$ | $0.4740_{\pm 0.0003}$ | $0.3919_{\pm 0.0019}$ | $0.2674_{\pm 0.0031}$ |
| GatedGCN + RRWP | $0.6516_{\pm 0.0072}$ | $0.2514_{\pm 0.0001}$ | - | - | - |
| GraphGPS + noPE | $0.6514_{\pm 0.0123}$ | $0.4243_{\pm 0.0305}$ | $0.4649_{\pm 0.0025}$ | $0.4517_{\pm 0.0112}$ | $0.3799_{\pm 0.0056}$ |
| GraphGPS + LapPE | $0.6620_{\pm 0.0073}$ | $0.2497_{\pm 0.0024}$ | $0.4696_{\pm 0.0017}$ | $0.4505_{\pm 0.0062}$ | $0.3859_{\pm 0.0016}$ |
| GraphGPS + ESLapPE | $0.6516_{\pm 0.0062}$ | $0.2568_{\pm 0.0013}$ | $0.4639_{\pm 0.0031}$ | $0.4538_{\pm 0.0083}$ | $0.3866_{\pm 0.0017}$ |
| GraphGPS + RWSE | $0.6510_{\pm 0.0071}$ | $0.2549_{\pm 0.0033}$ | $0.4685_{\pm 0.0009}$ | $0.4531_{\pm 0.0073}$ | $0.3891_{\pm 0.0033}$ |
| GraphGPS + SignNet | $0.5719_{\pm 0.0055}$ | $0.2657_{\pm 0.0021}$ | $0.4624_{\pm 0.0020}$ | $0.4291_{\pm 0.0053}$ | - |
| GraphGPS + GCKN | $0.6502_{\pm 0.0101}$ | $0.2519_{\pm 0.0009}$ | $0.4609_{\pm 0.0007}$ | $0.4515_{\pm 0.0010}$ | - |
| GraphGPS + WLPE | $0.5851_{\pm 0.0441}$ | $0.5203_{\pm 0.0504}$ | $0.4622_{\pm 0.0012}$ | $0.4501_{\pm 0.0057}$ | - |
| GraphGPS + RWDIFF | $0.6519_{\pm 0.0077}$ | $0.4769_{\pm 0.0360}$ | $0.4669_{\pm 0.0006}$ | $0.4488_{\pm 0.0097}$ | $0.3873_{\pm 0.0024}$ |
| GraphGPS + RRWP | $0.6505_{\pm 0.0058}$ | $0.3734_{\pm 0.0157}$ | - | - | - |
| Exphormer + noPE | $0.6200_{\pm 0.0052}$ | $0.2584_{\pm 0.0019}$ | $0.4661_{\pm 0.0021}$ | $0.4149_{\pm 0.0047}$ | $0.3445_{\pm 0.0052}$ |
| Exphormer + LapPE | $0.6424_{\pm 0.0063}$ | $0.2496_{\pm 0.0024}$ | $0.4737_{\pm 0.0024}$ | $0.4242_{\pm 0.0044}$ | $0.3471_{\pm 0.0028}$ |
| Exphormer + ESLapPE | $0.6281_{\pm 0.0085}$ | $0.2513_{\pm 0.0022}$ | $0.4676_{\pm 0.0018}$ | $0.4141_{\pm 0.0054}$ | $0.3485_{\pm 0.0011}$ |
| Exphormer + RWSE | $0.6240_{\pm 0.0069}$ | $0.2579_{\pm 0.0010}$ | $0.4642_{\pm 0.0039}$ | $0.4218_{\pm 0.0063}$ | $0.3485_{\pm 0.0011}$ |
| Exphormer + SignNet | $0.5458_{\pm 0.0097}$ | $0.2667_{\pm 0.0037}$ | $0.4615_{\pm 0.0066}$ | $0.3966_{\pm 0.0020}$ | - |
| Exphormer + GCKN | $0.6422_{\pm 0.0080}$ | $0.2514_{\pm 0.0012}$ | $0.4604_{\pm 0.0038}$ | $0.4196_{\pm 0.0049}$ | - |
| Exphormer + WLPE | $0.6216_{\pm 0.0069}$ | $0.2558_{\pm 0.0011}$ | $0.2051_{\pm 0.0080}$ | $0.4104_{\pm 0.0071}$ | - |
| Exphormer + RWDIFF | $0.6275_{\pm 0.0031}$ | $0.2556_{\pm 0.0021}$ | $0.4642_{\pm 0.0032}$ | $0.4165_{\pm 0.0059}$ | $0.3417_{\pm 0.0006}$ |
| Exphormer + RRWP | $0.6208_{\pm 0.0074}$ | $0.2586_{\pm 0.0014}$ | - | - | - |
| SparseGRIT + noPE | $0.4885_{\pm 0.0036}$ | $0.2550_{\pm 0.0006}$ | $0.4527_{\pm 0.0006}$ | $0.3471_{\pm 0.0030}$ | $0.1976_{\pm 0.0038}$ |
| SparseGRIT + LapPE | $0.5884_{\pm 0.0059}$ | $0.2487_{\pm 0.0014}$ | $0.4585_{\pm 0.0011}$ | $0.3514_{\pm 0.0026}$ | $0.1974_{\pm 0.0008}$ |
| SparseGRIT + ESLapPE | $0.5161_{\pm 0.0069}$ | $0.2537_{\pm 0.0005}$ | $0.4532_{\pm 0.0005}$ | $0.3462_{\pm 0.0035}$ | $0.1958_{\pm 0.0001}$ |
| SparseGRIT + RWSE | $0.5570_{\pm 0.0079}$ | $0.2537_{\pm 0.0012}$ | $0.4553_{\pm 0.0014}$ | $0.3460_{\pm 0.0071}$ | $0.1969_{\pm 0.0010}$ |
| SparseGRIT + SignNet | $0.5115_{\pm 0.0064}$ | $0.2640_{\pm 0.0018}$ | $0.4573_{\pm 0.0003}$ | $0.3419_{\pm 0.0014}$ | - |
| SparseGRIT + GCKN | $0.5871_{\pm 0.0042}$ | $0.2492_{\pm 0.0010}$ | $0.4500_{\pm 0.0004}$ | $0.3519_{\pm 0.0040}$ | - |
| SparseGRIT + WLPE | $0.4808_{\pm 0.0016}$ | $0.2547_{\pm 0.0005}$ | $0.4489_{\pm 0.0012}$ | $0.3439_{\pm 0.0027}$ | - |
| SparseGRIT + RWDIFF | $0.5521_{\pm 0.0072}$ | $0.2550_{\pm 0.0008}$ | $0.4551_{\pm 0.0005}$ | $0.3447_{\pm 0.0046}$ | $0.1965_{\pm 0.0011}$ |
| SparseGRIT + RRWP | $0.6702_{\pm 0.0080}$ | $0.2504_{\pm 0.0025}$ | - | - | - |
| GRIT + noPE | $0.4861_{\pm 0.0053}$ | $0.2489_{\pm 0.0008}$ | $0.4525 \pm 0.0001$ | $0.3556 \pm 0.0019$ | $0.2105 \pm 0.0004$ |
| GRIT + LapPE | $0.5834_{\pm 0.0105}$ | $0.2474_{\pm 0.0005}$ | $0.4580 \pm 0.0020$ | $0.3551 \pm 0.0032$ | $0.2112 \pm 0.0005$ |
| GRIT + ESLapPE | $0.4831_{\pm 0.0023}$ | $0.2584_{\pm 0.0002}$ | $0.4486 \pm 0.0014$ | $0.3485 \pm 0.0028$ | $0.2100 \pm 0.0008$ |
| GRIT + RWSE | $0.5432_{\pm 0.0034}$ | $0.2612_{\pm 0.0008}$ | $0.4524 \pm 0.0001$ | $0.3461 \pm 0.0058$ | $0.2114 \pm 0.0009$ |
| GRIT + SignNet | $0.5307_{\pm 0.0085}$ | $0.2600_{\pm 0.0018}$ | $0.4608 \pm 0.0007$ | $0.3385 \pm 0.0045$ | - |
| GRIT + GCKN | $0.5868_{\pm 0.0051}$ | $0.2477_{\pm 0.0006}$ | $0.4521 \pm 0.0002$ | $0.3516 \pm 0.0003$ | - |
| GRIT + WLPE | $0.4798_{\pm 0.0012}$ | $0.2578_{\pm 0.0011}$ | $0.4515 \pm 0.0004$ | $0.3441 \pm 0.0011$ | - |
| GRIT + RWDIFF | $0.5801_{\pm 0.0036}$ | $0.2639_{\pm 0.0010}$ | $0.4563 \pm 0.0003$ | $0.3521 \pm 0.0079$ | $0.2128 \pm 0.0008$ |
| GRIT + RRWP | $0.6865_{\pm 0.0050}$ | $0.2454_{\pm 0.0010}$ | - | - | - |

Table 20: Results for three OGB datasets.

| Sparse Graph | ogbn-arxiv | ogbg-molhiv | ogbg-molpcba |
|---|---|---|---|
| GPS + noPE | - | $77.885_{\pm 2.641}$ | $28.573_{\pm 0.215}$ |
| GPS + ESLapPE | - | $78.295_{\pm 0.925}$ | $28.373_{\pm 0.477}$ |
| GPS + LapPE | - | $77.256_{\pm 0.806}$ | $29.325_{\pm 0.300}$ |
| GPS + GCKN | - | $77.652_{\pm 1.326}$ | - |
| GPS + WLPE | - | $75.835_{\pm 0.857}$ | $27.968_{\pm 0.154}$ |
| GPS + RWSE | - | $77.890_{\pm 1.045}$ | $28.563_{\pm 0.283}$ |
| GPS + RRWP | - | $76.383_{\pm 1.189}$ | $28.765_{\pm 0.268}$ |
| Exphormer + noPE | $70.782_{\pm 0.029}$ | $78.347_{\pm 0.440}$ | $28.355_{\pm 0.224}$ |
| Exphormer + ESLapPE | - | $77.578_{\pm 1.595}$ | $28.123_{\pm 0.281}$ |
| Exphormer + LapPE | - | $76.818_{\pm 0.744}$ | $27.858_{\pm 0.082}$ |
| Exphormer + GCKN | - | $78.045_{\pm 1.146}$ | - |
| Exphormer + WLPE | $70.738_{\pm 0.095}$ | $75.587_{\pm 1.172}$ | $27.283_{\pm 0.312}$ |
| Exphormer + RWSE | $70.693_{\pm 0.132}$ | $77.053_{\pm 0.295}$ | $28.490_{\pm 0.257}$ |
| Exphormer + RRWP | - | $77.305_{\pm 1.250}$ | - |
| SparseGRIT + noPE | $70.955_{\pm 0.119}$ | $77.752_{\pm 1.331}$ | $20.950_{\pm 0.076}$ |
| SparseGRIT + ESLapPE | - | $77.670_{\pm 1.870}$ | $20.953_{\pm 0.086}$ |
| SparseGRIT + LapPE | - | $75.393_{\pm 1.358}$ | $22.748_{\pm 0.455}$ |
| SparseGRIT + GCKN | - | $75.453_{\pm 0.893}$ | - |
| SparseGRIT + WLPE | $70.877_{\pm 0.045}$ | $76.060_{\pm 1.066}$ | $20.670_{\pm 0.211}$ |
| SparseGRIT + RWSE | - | $76.973_{\pm 0.242}$ | $23.628_{\pm 0.205}$ |
| SparseGRIT + RRWP | - | $78.353_{\pm 0.546}$ | - |

Table 21: Running Time for Pretransforming PEs (measured in seconds) on BENCHMARKINGGNN

| | MNIST | CIFAR10 | PATTERN | CLUSTER | ZINC |
|---|---|---|---|---|---|
| LapPE | 93 | 123 | 28 | 23 | 8 |
| ESLapPE | 90 | 122 | 28 | 21 | 7 |
| RWSE | 225 | 252 | 55 | 53 | 31 |
| SignNet | 90 | 122 | 28 | 20 | 7 |
| PPR | 585 | 600 | 90 | 71 | 313 |
| GCKN | 1180 | 1705 | 552 | 448 | 89 |
| WLPE | 166 | 217 | 166 | 108 | 8 |
| RWDIFF | 210 | 209 | 226 | 158 | 15 |
| RRWP | 159 | - | 47 | 53 | 22 |

Table 22: Memory Usage for Pretransforming PEs (measured in MB) on BENCHMARKINGGNN

| | MNIST | CIFAR10 | PATTERN | CLUSTER | ZINC |
|---|---|---|---|---|---|
| LapPE | 795.55 | 1021.10 | 290.78 | 251.45 | 67.68 |
| ESLapPE | 652.55 | 809.55 | 293.79 | 377.49 | 148.95 |
| RWSE | 772.05 | 994.92 | 286.80 | 243.91 | 60.12 |
| SignNet | 803.17 | 1024.11 | 293.41 | 377.97 | 58.90 |
| PPR | 2430.63 | 4142.62 | 1849.49 | 1701.08 | 1041.20 |
| GCKN | 355.12 | 824.58 | 292.69 | 170.61 | 68.00 |
| WLPE | 1268.05 | 1709.76 | 1506.82 | 809.11 | 104.82 |
| RWDIFF | 264.00 | 312.49 | 145.37 | 191.87 | 28.38 |
| RRWP | 43327.52 | - | 29823.30 | 36577.61 | 2414.64 |

Table 23: Running Time for Pretransforming PEs (measured in seconds) on LRGB

| | Peptides-func | Peptides-struct | PCQM-Contact | PascalVOC-SP | COCO-SP |
|---|---|---|---|---|---|
| LapPE | 44 | 44 | 358 | 203 | 2889 |
| ESLapPE | 44 | 45 | 384 | 203 | 3045 |
| RWSE | 92 | 90 | 1500 | 418 | 8373 |
| SignNet | 44 | 44 | 386 | 206 | - |
| GCKN | 530 | 527 | 5072 | 2377 | - |
| WLPE | 34 | 33 | 451 | 97 | - |
| RWDIFF | 47 | 46 | 1075 | 764 | 8976 |

Table 24: Memory Usage for Pretransforming PEs (measured in MB) on LRGB

| | Peptides-func | Peptides-struct | PCQM-Contact | PascalVOC-SP | COCO-SP |
|---|---|---|---|---|---|
| LapPE | 438.33 | 651.95 | 4625.45 | 875.47 | 9149.31 |
| ESLapPE | 438.84 | 654.52 | 4617.98 | 877.84 | 9153.73 |
| RWSE | 424.32 | 421.45 | 4410.43 | 875.87 | 9112.50 |
| SignNet | 647.73 | 646.92 | 6091.65 | 871.27 | - |
| GCKN | 662.06 | 661.18 | 6016.68 | 909.13 | - |
| WLPE | 710.55 | 709.84 | 6525.88 | 1639.22 | - |
| RWDIFF | 334.18 | 334.54 | 3484.15 | 677.29 | 7248.34 |

Table 25: Running Time for Pretransforming PEs (measured in seconds) on OGB

| | ogbn-arxiv | ogbg-molhiv | ogbg-molpcba |
|---|---|---|---|
| ESLapPE | - | 28 | 296 |
| LapPE | - | 26 | 314 |
| GCKN | - | 325 | - |
| WLPE | 14 | 31 | 334 |
| RWSE | 35 | 101 | 1034 |
| RRWP | - | 54 | - |

Table 26: Memory Usage for Pretransforming PEs (measured in MB) on OGB

| | ogbn-arxiv | ogbg-molhiv | ogbg-molpcba |
|---|---|---|---|
| ESLapPE | - | 296.21 | 4374.86 |
| LapPE | - | 328.23 | 4514.98 |
| GCKN | - | 238.57 | - |
| WLPE | 57.76 | 382.05 | 5080.64 |
| RWSE | - | 301.18 | 2799.23 |
| RRWP | - | 3220.03 | - |