# OpenReview forum: "Benchmarking Positional Encodings for GNNs and Graph Transformers"
_ICLR.cc/2025/Conference — Submitted to ICLR 2025_

### Official Review · Reviewer_Uhj9 · 2024-10-25

**Soundness:** 4
**Presentation:** 4
**Contribution:** 2
**Rating:** 5
**Confidence:** 4

**Summary:**

This paper provides a comprehensive benchmarking study on the effectiveness of various PEs for GNNs within a unified framework -- it investigates the impact of PEs on several types of models, including MPNNs and GTs, benchmarking the performance. The paper introduces a modified attention mechanism based on GRIT, denoted SparseGRIT, which facilitates localised graph convolutions through a sparsified attention mechanism.

**Strengths:**

1. **Comprehensive Benchmarking**: The study evaluates multiple PEs across MPNNs and GTs on several datasets, providing useful comparisons.
2. **Reproducibility**: A code for reproducing the results is provided.
3. **Clarity**: This paper is well written and easy to follow.

**Weaknesses:**

1. **Limited Novelty**: The paper mainly focuses on benchmarking PEs across various architectures, rather than introducing fundamentally new models or methodologies (beyond SparseGRIT, which is a straightforward extension of GRIT). While benchmarking is important, the lack of novelty limits the impact of the paper.

2. **Justification for PE Performance**: While the authors effectively demonstrate empirical differences in the performance of PEs across different models, they do not provide a theoretical explanation for the observed variations. Providing a theoretical explanation or justification to complement the empirical findings would enhance the depth and impact of the paper. For example, showing that certain PEs are more expressive (for example in graph separation power) than others would benefit the paper. Another option is to show and analyze empirically what structural information is captured via the different PEs.

3. **Marginal Performance Gains**: The benchmarking results reveal no significant performance improvements over existing methods, except for the PCQM-Contact dataset. This raises concerns about the practical impact of the paper beyond its benchmarking contribution.

**Questions:**

Do you have any theoretical justifications to support the empirical results observed for the different positional encodings across the different models? A theoretical perspective that could help explain why specific PEs outperform others, and in which cases, would improve the quality of the paper.

---

> ### Author Response · Authors · 2024-11-17
>
> We thank the reviewer for their thoughtful feedback and valuable suggestions.
>
> **W1: Limited Novelty**
>
> We appreciate the reviewer's recognition of the value of our comprehensive benchmarking study. While it is true that our main focus is on systematically evaluating PEs across different GNN architectures, we believe that this type of empirical analysis is highly valuable to the community, especially given the rapid proliferation of PEs in recent years. Our benchmarking framework is designed to fill a gap in the current landscape by providing a unified evaluation across multiple architectures, including MPNNs and GTs.
> Regarding SparseGRIT, we acknowledge that it is an extension of GRIT; however, this approach has not been thoroughly explored in prior works, especially within the context of benchmarking PEs, and offers a practical contribution to making GTs more scalable.
>
> **W2: Justification for PE Performance**
>
> We appreciate the reviewer's suggestion for a deeper theoretical analysis of PE performance. In response, we have added a new section titled “Guidelines for Practitioners” to provide more concrete recommendations on selecting PEs based on our findings. Our study focuses on real-world datasets that are commonly used for benchmarking, where it is challenging to directly correlate the theoretical expressiveness of PEs with practical performance. We believe that our empirical results, alongside the new guidelines, can serve as a valuable resource for researchers and practitioners in choosing the most suitable PEs for their specific tasks.
>
> **W3: Marginal Performance Gains**
>
> Our primary goal was not to propose new state-of-the-art models but to provide a thorough benchmarking of PEs across a diverse set of GNN architectures. A key takeaway from our study is that older models leveraging sparse graph topology can achieve performance on par with more complex, recent architectures when the right PE is applied. This is a valuable insight, showing that the choice of PE can be as impactful as architectural advancements. While significant improvements were not the primary focus, our benchmarking results on datasets like PCQM-Contact, where we set new state-of-the-art performance, highlight the potential of optimizing PEs. Notably, using models like Exphormer, we achieved substantial gains across several datasets, demonstrating that well-chosen PEs can unlock performance without the need for overly complex models.
>
> **Q1: Theoretical Justification for Empirical Results**
>
> Our study primarily focuses on empirical evaluations using real-world datasets that are commonly used for benchmarking. It is often challenging to establish a direct connection between the theoretical expressiveness of positional encodings and their practical performance on these datasets. Real-world graphs come with inherent complexities and noise, which can obscure the direct impact of theoretical properties. Nevertheless, we reference recent work that suggests some PEs have higher graph separation power. While these insights can offer some context, our primary aim was to provide empirical guidance for practitioners based on observed results across diverse, real-world scenarios that are used to evaluate new state-of-the-art models.

---

> > ### Comment · Reviewer_Uhj9 · 2024-11-20
> >
> > I thank the authors for the rebuttal and detailed response. I would like to keep my current score.

---

### Official Review · Reviewer_nezT · 2024-10-27

**Soundness:** 3
**Presentation:** 3
**Contribution:** 2
**Rating:** 5
**Confidence:** 4

**Summary:**

The paper presents a benchmark study of positional encodings (PEs) in graph neural networks and graph transformers. The work encompasses three contributions: (1) the main one is a systematic evaluation of various PEs across different architectures, examining multiple PE methods across several benchmark datasets including very popular ones. This is accompanied by two other contributions: (2) theoretical insights connecting Graph Transformers (GTs) and the Weisfeiler-Lehman (WL) test, proving that GTs can be bounded by the 1-WL algorithm on fully connected graphs (can certain GTs can reach this bound), and (3) the introduction of SparseGRIT, a variant of GRIT that uses sparse connectivity while maintaining competitive performance.



Conclusion:
While the paper addresses an important practical need and provides valuable empirical insights, its contributions fall short of the novelty and depth expected at ICLR. The work would be more suitable for a venue focused on empirical machine learning studies or as a journal submission with expanded analysis. The authors should consider deepening their theoretical analysis, providing more interesting distinctions between different types of encodings before submitting to a more appropriate venue.

**Strengths:**

The paper addresses a significant practical need in the field, as the growth of PE schemes in recent years has made it challenging for practitioners to choose appropriate methods for their tasks. The authors examine multiple different PE methods across multiple architectures, providing valuable insights into their relative performance. Their comprehensive evaluation reveals that previously untested combinations can outperform existing methods. The authors provide a clear presentation of results including standard deviations across multiple runs, with detailed runtime and memory analysis for each PE method. The reproducibility of the work is strengthened by the authors' release of a unified codebase implementing all tested models and encodings.

**Weaknesses:**

The theoretical contributions of the paper are relatively modest. The classification of PEs into Laplacian-based, random walk-based, and others (Section 3.1) lacks depth, merely organizing existing methods without providing new insights into their relationships. The theoretical connection between GTs and WL test presented in Section 3.4 largely follows from existing work, with Theorem 3.2 being a straightforward extension of previous results.

The paper would benefit from a clearer distinction between structural and positional encodings, as understanding their different roles is crucial for practitioners [see "On the Equivalence between Positional Node Embeddings and Structural Graph Representations" by Srinivasan and Ribeiro and followup papers]. The evaluation ignores this important distinction, treating all encodings as equivalent augmentations to node features.

The presentation of SparseGRIT as a novel contribution requires more justification. The original GRIT paper may have already employed sparse variants in certain cases, and the paper would benefit from a more detailed discussion of how SparseGRIT differs from existing GRIT implementations beyond the simplified attention mechanism described in Equation 1.

**Questions:**

The authors should clarify how SparseGRIT differs from sparse implementations in the original GRIT paper.

---

> ### Author Response · Authors · 2024-11-17
>
> We thank the reviewer for their feedback.
>
> **Clarification on SparseGRIT**
>
> We appreciate the reviewer's question regarding the distinction between our SparseGRIT implementation and potential sparse variants in the original GRIT paper. Unfortunately, the original paper does not provide additional analysis or a formal definition for sparse implementations, and even the GRIT codebase lacks the necessary components to reproduce their results. Based on our evaluation, we assume similarities, as the performance on the ZINC dataset (the only dataset where they provide results for this) appears comparable. However, due to the absence of reproducible details, a direct comparison remains challenging.
>
> **Response to Theoretical Contributions and Classification of PEs**
>
> We acknowledge the reviewer's feedback on the theoretical aspects of our work. The categorization of PEs into Laplacian-based, random walk-based, and other methods was intentionally designed to reflect the distinct performance trends observed in our empirical results. These distinctions have been further elaborated in our newly added section, “Guidelines for Practitioners.” While our primary focus is on benchmarking, these categories can help practitioners to choose a suitable PE.
> Regarding the suggestion to differentiate between structural and positional encodings, we opted not to pursue this distinction in our analysis, as our empirical findings did not reveal practical implications that would benefit from such categorization. We aimed to keep our study focused on actionable insights rather than theoretical distinctions that do not translate into observable performance differences.
>
> **Response to Venue Suitability and Empirical Focus**
>
> We understand the reviewer's concern about the balance between theoretical and empirical contributions. However, we respectfully disagree with the notion that our work may not be suitable for ICLR. As the reviewer acknowledges, our study addresses a significant practical need by providing a systematic evaluation of PEs, which has been a growing challenge for practitioners in the field. The ICLR call for papers explicitly welcomes “Datasets and Benchmarks” submissions, and recent ICLR publications include several studies of a purely empirical nature. Given that our paper aims to fill a critical gap in the benchmarking of PEs, we believe it aligns well with the scope of ICLR. We kindly request the reviewer to consider the practical value of our contributions and the relevance of empirical studies to the ICLR community.

---

> > ### Comment · Reviewer_nezT · 2024-11-25
> >
> > I thank the authors for their response. I believe the paper addresses an important issue considering the various positional and structural encodings used today. It is also clear that the authors worked very hard on this paper. Nevertheless, in its current form and in my opinion, the paper is not strong enough to be accepted to ICLR. I believe the authors should continue working in this direction and augment the current version with more contributions. I will raise my score to support the authors.

---

### Official Review · Reviewer_6s7w · 2024-11-02

**Soundness:** 3
**Presentation:** 2
**Contribution:** 2
**Rating:** 5
**Confidence:** 4

**Summary:**

This paper aims to systematically evaluate positional encodings (PEs) across various graph neural networks (GNNs) and graph transformer (GT) architectures. Positional encodings are critical for retaining graph topological information in GTs. The authors propose a unified benchmarking framework that isolates the effects of PEs from architectural innovations and find that previous untested combinations of PEs and architectures can lead to SOTA performance. The also explores the connection between message-passing networks and transformers and introduce a sparsified GRIT attention mechanism to balance local and global information.

**Strengths:**

1. The authors propose a unified benchmark, allowing for a comprehensive evaluation of PEs independent of architecture. They reproduces state-of-the-art MPNNs as well as GTs and integrate them into their codebase for comparison.
2. The experiments seem solid and provide insights into difference combinations of architectures and PEs.
3. The authors provide extensive details on their experiment setups and results, which is good for reproduction.

**Weaknesses:**

1. The authors notes significant memory and computational demands for some random walk-based PEs (e.g., RRWP), which limits their practicality on large graphs. Could the authors provide more experiment results about time and space complexity?
2. The authors may explore additional graph data types to confirm the consistency of PE impacts across different tasks beyond current settings.
3. The authors may include more clarification and explanations on PEs in the main text, which may be challenging for readers unfamiliar with advanced PE concepts.

**Questions:**

1. The authors mention that the best PE choice varies depending on tasks and datasets, and sometimes the model can achieve good performance without any PE. Can the authors clarify more clearly why this happens or why PE doesn't work well sometimes? Are the existing PE designs not suitable for the particular case or it doesn't need any PE indeed?
2. The author provides their code but I couldn't open it. I don't know whether this is really a problem or is due to the network of my computer.
3. For node-level tasks the author include arxiv dataset. Could the author evaluate the PE designs on a larger-scale dataset that can compare their performance on large graphs? Can the authors provide more insights into the scalabilid GRIT mechanism when applied to very large-scale graphs?

---

> ### Author Response · Authors · 2024-11-17
>
> We thank the reviewer for their detailed feedback and constructive suggestions.
>
> **W1**
>
> We provide detailed analyses of time and space requirements in Tables 21 to 26 and Figure 6 in the Appendix, alongside a discussion in the main text. These sections specifically address the computational demands of various PEs, including RRWP, on large graphs.
>
> **W2**
>
> Our extended experiments include results on three OGB datasets, detailed in the Appendix. However, due to computational resource constraints, it is challenging to run exhaustive comparisons across all possible settings, as this would require an extensive number of runs. The datasets highlighted in the main text are widely recognized benchmarks for evaluating new architectures, which makes them highly relevant for our analysis. Additionally, our benchmarking framework is designed to be extendable and allows researchers to incorporate and evaluate new PEs and GNN models as they become available.
>
> **W3**
>
> The main text includes an introduction to PEs, with comprehensive details provided in the Appendix. If there are specific aspects of PEs that require further clarification, we would greatly appreciate more targeted feedback on what additional explanations might be beneficial for readers less familiar with these concepts.
>
> **Q1**
>
> As our evaluation primarily focuses on real-world graph datasets, it is challenging to pinpoint why certain PEs outperform others, as their effectiveness is highly task-dependent. We have added a detailed discussion on this topic in the newly introduced section, “Guidelines for Practitioners,” which outlines scenarios where different PEs are likely to be most effective.
>
> **Q2**
>
> The code was intended to be accessible throughout the entire review period. We have verified its availability on our end and are not experiencing any access issues. Could you confirm if this problem persists for you? We are happy to assist in troubleshooting any access difficulties.
>
> **Q3**
>
> As discussed in our paper, the GRIT mechanism is limited in scalability due to its computational complexity, which hinders its applicability to very large graphs. However, we provide results for more scalable combinations in the Appendix, which demonstrate how various methods perform on larger datasets.

---

> > ### Comment · Reviewer_6s7w · 2024-11-25
> >
> > I thank the authors for their responses but I would like to maintain my score.

---

### Official Review · Reviewer_M4w4 · 2024-11-02

**Soundness:** 2
**Presentation:** 3
**Contribution:** 1
**Rating:** 5
**Confidence:** 4

**Summary:**

This paper introduces a unified evaluation framework and empirically investigates various combinations of positional encodings (PEs) and GNNs/GTs across multiple datasets, uncovering new combinations that surpass the current state-of-the-art. The results show that PEs can significantly impact model performance depending on the task, though they may sometimes reduce effectiveness. Additionally, the paper provides a theoretical analysis of the relationship between MPNNs and GTs using the WL test, and proposes a sparse MPNN architecture based on GRIT.

**Strengths:**

1. The paper is well-written and presented, with both theoretical and empirical analyses structured and clearly explained.

2. The extensive evaluation of PE and GNN/GT combinations across diverse datasets is commendable, establishing new state-of-the-art results on two sets of benchmark datasets.

3. The codebase, which includes implementations of all models and PEs, is a valuable resource for future researchers.

**Weaknesses:**

The primary limitation of this paper is its contribution.
1. Although the paper provides a valuable benchmarking evaluation on an important problem—understanding which PEs work best for specific tasks and architectures—the analysis offers limited new insights. Previous work has shown that different PEs yield different performance improvements based on the task. It would be more valuable if the authors provided deeper insights into this phenomenon, for instance, by characterizing tasks to explain why certain PEs (e.g., Laplacian-based, Random-walk-based, or others) are more beneficial. Even a theoretical analysis or synthetic tasks exploring this could help future researchers select the most suitable PEs based on task requirements.
2. Another contribution of the paper is the theoretical analysis of the relationship between MPNNs and GTs using the 1-WL test, interpreting GTs as message-passing on a new topology. However, the theorem’s significance is limited, as it is well-known that GTs with full attention can be seen as MPNNs on a fully-connected graph. Additionally, the theorem’s bounds on GT expressiveness based on the 1-WL test apply to the dependency graph rather than the original graph. The latter would have a more direct effect on downstream performance.

**Questions:**

1. How do the empirical results relate to theoretical analyses of PEs, such as [1]?
2. Were pretrained PEs, such as GPSE [2], considered?
3. Why are prediction heads included in the design space, given that they are typically task-specific?
4. Why was GRIT chosen as the basis for proposing a sparse version of GTs? Could this approach be generalized to other GTs?

[1] Comparing Graph Transformers via Positional Encodings. Black et al., ICML 2024.
[2] Graph Positional and Structural Encoder. Cantrürk et al., ICML 2024.

---

> ### Author Response · Authors · 2024-11-17
>
> We thank the reviewer for their detailed feedback and constructive suggestions.
>
> **W1**
>
> Thank you for the suggestion. In response, we have added a new section titled “Guidelines for Practitioners” to the paper to provide more concrete recommendations based on our findings.
>
> **W2**
>
> Our theorem demonstrates that GTs can achieve the same expressiveness as MPNNs when applied to a fully-connected graph. We provide a formal proof of this equivalence and outline the specific conditions under which it holds. This theoretical analysis is included to justify our use of fully-connected MPNNs in comparison with GTs, as well as to motivate the sparsification of GTs in order to isolate the impact of the convolutional update mechanism from the graph topology. While we acknowledge that this result may not be surprising, it is not intended to be the primary contribution of our paper. Rather, our main focus lies in the comprehensive benchmarking analysis of PEs across different architectures.
>
> **Q1**
>
> The categorization of PEs in [1] does not directly align with our approach, which focuses on empirical performance across diverse real-world datasets. We found that the theoretical expressiveness and categorization of PEs do not always translate to practical performance gains in these settings. Our goal was to benchmark the impact of different PEs on commonly used benchmarking datasets, where the theoretical power of PEs is not always indicative of their real-world effectiveness. As a result, we could not derive meaningful insights by directly comparing our empirical findings to the theoretical framework in [1].
>
> **Q2**
>
> We included a diverse set of PEs in our evaluation but were constrained by computational resources, which limited our ability to test every existing PE. Therefore, we prioritized evaluating the best-performing PEs and GT architectures based on prior literature and empirical performance. We believe this selection strikes a balance between comprehensiveness and feasibility, given the computational limits.
>
> **Q3**
>
> We included prediction heads in the design space due to their significant influence on model performance. While optimal configurations may vary per task, our primary focus was on understanding PE impacts within a consistent architecture.
>
> **Q4**
>
> We chose GRIT for its status as a state-of-the-art GT model leveraging full attention, which makes it particularly suitable for analyzing the effectiveness of its update mechanism in both dense and sparsified settings. This evaluation can also be extended to other GT models.

---

> > ### Comment · Reviewer_M4w4 · 2024-11-25
> >
> > Thank you for the detailed response. I appreciate the added "Guidelines for Practitioners", which enhances the contribution of the paper. I have adjusted the score in response to that.

---

### Official Review · Reviewer_znMq · 2024-11-03

**Soundness:** 2
**Presentation:** 2
**Contribution:** 1
**Rating:** 3
**Confidence:** 4

**Summary:**

The paper studies the effect of different positional encodings across different graph transformer (GT) architectures. The goal of the paper is to understand their effect by isolating their contribution, instead of presenting the observed performance improvements that can be attributed to both the positional encoding technique and the specific architecture used. They also discuss a connection between MPNNs and GTs, although this connection is well-established in the literature, and proposed a sparsified variant of GRIT.

**Strengths:**

The goal of the paper (benchmarking positional encodings by isolating their contribution) is very clear at the beginning of the paper, although it gets lost while the paper progresses (see weaknesses). This goal is important, and therefore the research question is significant.

**Weaknesses:**

1. The paper lacks focus. While it starts by discussing the goal as understanding the effect of different position encodings, everything before the evaluation section does not discuss this point sufficiently. On the contrary, the paper presents the connection between MPNNs and GTs as a contribution, despite being well known in the literature (see for instance Velickovic 2023) and not central to the main focus of the paper.

2. Because of the lack of focus, the contributions of the paper related to the main goal are limited. The paper only tests few positional encoding techniques and few graph transformers. More importantly, *there is no discussion on the conclusions of the benchmarking study*: which PE is the best under which assumptions? The authors only discuss which PE is the best based on the dataset. I think it is necessary to have a section discussing when it is best to use a certain PE, based on the characteristics of the dataset (e.g., large graphs, local/global information needed for the prediction), of the task, or of the architecture used. Otherwise the paper does not help practitioners in their choice of PE, it simply lists which one is best for that particular dataset tested, and any variation in the dataset could lead to completely different conclusions.

Velickovic 2023. Everything is Connected: Graph Neural Networks

**Questions:**

1. Can you draw explicit conclusions about your findings? Specifically, please clarify conditions under which a particular PE may outperform others. I recommend that the authors include a specific section analyzing patterns in their results - for example, do certain types of PEs consistently perform better on larger graphs or tasks requiring more global information? This would help practitioners apply the findings more broadly beyond just the specific datasets tested.

2. I would encourage also to review the literature. For instance, the connection between MPNNs and GTs has been extensively studied (e.g., Cai et al., 2023, Müller et al., 2023) and you are missing a lot of related work. Also the introduction is missing citations to support your claims.

3. If you are benchmarking PE, why the tables report the best PE for each model? This naturally defeats the purpose of benchmarking PE, as you are only showing the best PE for that dataset and architecture.  Can you include tables or figures showing the performance of all PE-model combinations?


Cai et al., 2023. On the Connection Between MPNN and Graph Transformer

Müller et al., 2024. Aligning Transformers with Weisfeiler-Leman

**Details Of Ethics Concerns:**

No ethics concerns

---

> ### Author Response · Authors · 2024-11-17
>
> We thank the reviewer for their insightful feedback.
>
> **W1**
>
> We appreciate the reviewer pointing out the missing references, which we have now included in the updated manuscript. These references were indeed an oversight in the original submission, and we regret their omission. We have also clarified our contribution more explicitly. While it is known that MPNNs and Graph Transformers share certain connections, our analysis focuses on a specific equivalence: we formally show that a GT is equally expressive to an MPNN on a fully-connected graph, independent of the PE used. We provide specific conditions under which this equivalence holds and demonstrate cases where certain GTs, such as Exphormer, face limitations, while GRIT is more expressive. Although this theoretical analysis is not our primary contribution, it provides valuable insight into why GRIT might outperform other methods in our empirical analysis. It also motivates the use of fully-connected MPNNs as a comparison to GTs and justifies the sparsification of attention mechanisms in GTs, an angle that has not been fully explored before. If you believe our analysis has been previously established, we kindly ask for a reference to support this claim.
>
> **W2**
>
> We aimed to include a comprehensive selection of state-of-the-art models and PEs. However, due to computational limitations, it was not feasible to test every existing method. This is why we provide a benchmarking framework and an extensible codebase, allowing researchers to easily include new PEs and GNN architectures. Regarding specific guidelines, we acknowledge that these were not clearly articulated in the original submission; hence, we have added a dedicated section to address them explicitly.
>
> **Q1**
>
> As mentioned in response to W2, we have added a new section to outline clear guidelines for practitioners.
>
> **Q2**
>
> We have already addressed this concern in W1. The missing references have been added, but we reiterate that our findings offer contributions that differ from previous work.
>
> **Q3**
>
> The results you requested are provided in Figure 4 and 5 of the Appendix, with corresponding tables in Table 18 and 19. These are all referenced in the main text. In the main manuscript, we focus on two primary perspectives: (1) comparing how PEs perform on various datasets (Figure 2 and 3), and (2) determining the best overall results achievable (Table 1 and 2). The latter was intended to present a new perspective on the current state-of-the-art, which we believe is a significant contribution. Specifically, we demonstrate that older architectures like Exphormer can perform on par with or even outperform the latest state-of-the-art models when paired with modern PEs. We consider this finding crucial to highlight in the main paper.

---

> > ### Comment · Reviewer_znMq · 2024-11-25
> >
> > I thank the authors for their replies but I also think the paper is not yet ready to be accepted. As I said in the original review, in my opinion, the presentation of the results is counterintuitive for a paper that aims at benchmarking PEs. This is because reporting only the best PE for a given architecture does not present the results necessary for drawing conclusions, which instead follow by looking at how a given PE performs across architectures. I would encourage the authors to rethink the structure of the paper and the presentations of the results, as the research direction is valuable.

---

### Meta-Review · Area_Chair_YcGZ · 2024-12-20

**Metareview:**

This paper presents a unified benchmark of PEs for MPNNs and GTs, with theoretical connections and a sparsified GRIT attention mechanism to explore global connectivity. The results show that new GNN-PE combinations outperform existing methods, offering a broader view of the state-of-the-art. Code is provided to support future research.

### Strengths:

1. The authors propose a unified benchmark that enables a comprehensive evaluation of PEs independent of architecture.
2. Results are clearly presented, including standard deviations across multiple runs, with detailed runtime and memory analysis for each PE method.
3. The provided codebase, including implementations of all models and PEs, is a valuable resource for future research.

### Weaknesses:

1. The theoretical contributions are relatively modest.
2. The paper would benefit from a clearer distinction between structural and positional encodings.
3. The paper tests only a few positional encoding techniques and graph transformers.
4. While the benchmarking evaluation is valuable for understanding which PEs work best for specific tasks and architectures, the analysis offers limited new insights.
5. The presentation could be improved by focusing more on the paper’s main goal, providing clearer explanations of PEs in the main text, and offering more justification for the presentation of SparseGRIT.

### Overall:

While this paper provides valuable contributions, it exhibits notable issues regarding theoretical novelty, experimental completeness, and clarity. A rejection is therefore recommended.

**Additional Comments On Reviewer Discussion:**

During the rebuttal period, the authors addressed the reviewers' concerns regarding presentation, theoretical analyses, and experimental evaluations. While most reviewers appreciated the revisions, particularly the added section "Guidelines for Practitioners", key concerns remain. Specifically, Reviewer znMq emphasized that the paper is not yet ready for acceptance, given the counterintuitive presentation of results for a benchmarking paper. Reviewer nezT also noted that the current version does not meet the strength required for acceptance to ICLR.

---

### Decision · Program_Chairs · 2025-01-22

Reject